# Exact Travelling-Wave Solutions of the Extended Fifth-Order Korteweg-de Vries Equation via Simple Equations Method (SEsM): The Case of Two Simple Equations

**DOI:** 10.3390/e24091288

**Published:** 2022-09-13

**Authors:** Elena V. Nikolova

**Affiliations:** 1Institute of Mechanics, Bulgarian Academy of Sciences, Acad. G. Bonchev Str., bl. 4, 1113 Sofia, Bulgaria; elena@imbm.bas.bg; Tel.: +359-02-979-64-43; 2Climate, Atmosphere and Water Research Institute, Bulgarian Academy of Sciences, Blvd. Tzarigradsko Chaussee 66, 1784 Sofia, Bulgaria

**Keywords:** extended fifth-order Korteweg-de Vries equation, simple equations method, exact travelling-wave solutions, composite functions

## Abstract

We apply the Simple Equations Method (SEsM) for obtaining exact travelling-wave solutions of the extended fifth-order Korteweg-de Vries (KdV) equation. We present the solution of this equation as a composite function of two functions of two independent variables. The two composing functions are constructed as finite series of the solutions of two simple equations. For our convenience, we express these solutions by special functions *V*, which are solutions of appropriate ordinary differential equations, containing polynomial non-linearity. Various specific cases of the use of the special functions *V* are presented depending on the highest degrees of the polynomials of the used simple equations. We choose the simple equations used for this study to be ordinary differential equations of first order. Based on this choice, we obtain various travelling-wave solutions of the studied equation based on the solutions of appropriate ordinary differential equations, such as the Bernoulli equation, the Abel equation of first kind, the Riccati equation, the extended tanh-function equation and the linear equation.

## 1. Introduction

Almost all processes occurring in human life and in nature can be considered to be complex systems. Examples of such complex systems are stock markets, research groups, traffic networks, etc. [1,2,3,4,5,6]. Moreover, most complex systems are characterized by their non-linearity. Examples of non-linear complex systems can be found in many scientific areas, from fluid mechanics and solid-state physics to biology and medicine [7,8,9,10,11]. Usually, the non-linear behavior of the complex systems is described by differential or difference equations [12,13,14,15]. In this direction, finding analytical and numerical solutions of non-linear differential equations is a great challenge for researchers from various scientific fields.

Research related to finding exact analytical solutions of non-linear partial differential equations (NPDFs) has a long history. At the beginning, to remove the non-linearity of the solved equation, an appropriate transformation is introduced. An example can be given the by so-called Hopf–Cole transformation [16,17], by which the non-linear Burger’s equation is reduced to the linear heat equation. Later, the transformation, which reduces the standard KdV equation to the famous linear equation of Schrdinger, leads to the appearance of the Method of Inverse Scattering Transform [18,19,20]. Other popular methods using appropriate transformations are the method of Hirota [21,22,23] and the method including the Painleve expansions [24,25,26].

In this study, we shall use the SEsM (Simple Equations Method) for obtaining exact solutions of non-linear differential equations. The idea for development of this method comes from the Method of Simplest Equation (MSE), proposed by Kudryashov [27]. MSE is based on searching for particular solutions of NPDEs as a series containing powers of solutions of a simpler equation called the simplest equation [28,29,30,31,32]. Application of the MSE for obtaining exact solutions of various evolution equations can be found in [33,34,35,36,37,38,39,40,41,42].

Returning to the methodology used in this study, we note that some ideas of SEsM were used in the papers of Martinov and Vitanov [43,44,45,46,47] as well as in the papers of Vitanov [48,49] about 30 years ago. About 10 years ago, Vitanov and co-authors developed methodology called Modified Method of Simplest Equation (MMSE) [50,51,52,53]. The MMSE was first applied for obtaining exact solutions of models in ecology and population dynamics [54,55,56]. In these investigations, the authors used the ordinary differential equation of Bernoulli as the simplest equation. Indeed, the main idea of the MMSE is the introduction of a balance equation. This equation allows the determination of the form of the solution of the solved equation as a finite series of the solution of the simplest equation. Moreover, it allows the determination of the kind of the simplest equations too. We note that with respect to presentation of the solution of the solved NPDE as a series of solution of the simplest equation by means of a balance equation, the MMSE is identical to the MSE developed by Kudryashov. Applications of MMSE for obtaining exact solutions of different non-linear differential equations can be found in [57,58,59,60,61,62,63,64,65,66,67,68]. In [57,58,59,60,61,62,63,64,65], the authors used ordinary differential equations of first order, such as the equation of Riccati, the equation of Bernoulli and the equation of Abel of the first kind as simplest equations. Ordinary differential equations of the second order, such as elliptic equations and equations, based on the function $1/\cosh{\left(\alpha x+\beta t\right)}^{n}$ are used as the simplest equations in [66,67,68].

In the last few years, Vitanov extended the MMSE to the SEsM [69,70]. In this extended version of the methodology, the solution of the solved NPDE is constructed as a composite function of the solutions of more simple equations. Moreover, the first step of the algorithm of SEsM includes introduction of an appropriate transformation, which allows the solved NPDE containing non-polynomial non-linearity to reduce to a NPDE, containing polynomial non-linearity [71,72]. Such a procedure allows the further application of the SEsM. In this direction, the SEsM covers all previous methodologies for finding exact solutions of PDEs to this end, as proved in [73,74,75,76,77,78]. Although the SEsM is a relatively new methodology, its application for finding exact solutions of different NPDEs can be seen in [79,80,81,82,83].

In this study, we shall focus on obtaining exact travelling-wave solutions of the extended fifth-order KdV equation. It is well known that the standard KdV equation is a general model for investigation of weakly non-linear long waves, including non-linearity and dispersion effects. In more detail, it was derived using a multi-scale asymptotic procedure on the governing Euler equations for inviscid and incompressible fluids, and primarily it described surface waves with long wavelength and small amplitude in shallow water [84] and internal waves in a shallow density-stratified fluid [85]. In fact, the KdV equation is obtained at a first-perturbation expansion (non-linearity and dispersion of first order are only taken into account). However, in many cases, the explanation of physical processes needs better precision. Then the influence of non-linear and dispersive terms with higher order in the physical systems cannot be neglected. In this case, applying the perturbation procedure to the governing Euler equations and leaving second-order terms in the perturbation expansions leads to the fifth-order KdV equation. In the context of propagating surface water waves, the fifth-order KdV equation was first proposed by Olver to describe the wave breaking [86]. Later, Marchant and Smyth [87] use the same equation to model more precisely the resonant flow of a fluid over topography. An equation of such a type was also derived in [88] to examine higher-order solitary-wave interactions. In [89], the author was derived the same equation to explain the surface waves in shallow water subjected to a linear shear flow. In the context of propagating internal waves in stratified media, the fifth-order KdV equation was proposed first by Koop and Butler [90] for a two-layer system, and then by Lamb and Yan [91] for a continuous density stratification with no free surface and without a basic shear flow. Next, the same equation was adapted by Pelinovsky et al. [92] to include a basic shear flow, but again with no free surface. Internal solitary waves in a stratified shear flow but with a free surface are modeled in [93] by the same evolution equation, as the authors expressed the model coefficients in terms of integrals of the modal function for the linear long-wave theory. In addition, the fifth-order KdV equation was used to describe internal waves of moderate amplitude in density-stratified fluids [94]. All these references are only a part of the possibilities that the studied equation gives in a purely physical sense. This emphasizes the importance of finding its exact analytical solutions.

The paper is structured as follows. In Section 2, we formulate the problem studied. The methodology of SEsM is presented in the same section. In Section 3, we present various types of exact travelling-wave solution of the extended fifth-order KdV equation depending on the simple equations used. Numerical examples of the obtained analytical solutions are shown in the same section. Some concluding remarks are made in Section 4.

## 2. Problem Formulation and Methodology

In this study, we discuss the extended KdV equation, presented in the form [86,87,88,89,90,91,92,93,94]:(1)$$u_{t}=u_{xxxxx}+\alpha uu_{xxx}-\beta u_{x}u_{xx}+\gamma u^{2}u_{x}+\delta u_{xxx}+\epsilon uu_{x}$$
where u(x,t) is a displacement of surface at any varied natural instances, *x* is the spatial coordinate, and *t* is time. In more detail, Equation (Equation 1) is a hydrodynamic model of an incompressible, inviscid fluid and its irrotational motion is governed by gravitational forces. In addition to the standard non-linear term with a coefficient ϵ=a/h<<1, and the standard linear dispersion term with a coefficient $\delta ={\left(h/l\right)}^{2}$<<1 (*a* denotes the wave amplitude, *h* the average depth of the fluid container, and *l* is the average wavelength), involved in the standard KdV equation, Equation (Equation 1) involves a cubic non-linear term (with a coefficient γ), a linear dispersion term of 5th order (with a coefficient of 1), and also higher-order non-linear dispersion terms with coefficients α and β.

Here, we shall search for analytical solutions of Equation (Equation 1) applying the SEsM. The SEsM can be used for obtaining analytical solutions of NPDEs:(2)$$\Phi \left(u(x,t),\ldots \ldots \ldots \right)=0$$
where the left-hand side of Equation (Equation 2) is a relationship containing the function u(x,t) and some of its derivatives.

The algorithm of SEsM includes the following four steps [71,72]:

(1). The transformation
(3)$$u\left(x,\ldots ,t\right)=Tr(F_{1}\left(x,\ldots ,t\right),F_{2}\left(x,\ldots ,t\right),\ldots F_{N}\left(x,\ldots ,t\right))$$
is made, where $Tr(F_{1}\left(x,\ldots ,t\right),F_{2}\left(x,\ldots ,t\right),\ldots F_{N}\left(x,\ldots ,t\right))$ is a composite function of other functions $F_{i}\;i=1\ldots N$. $F_{1}\left(x,\ldots ,t\right),F_{2}\left(x,\ldots ,t\right),\ldots ,F_{N}\left(x,\ldots ,t\right)$ are functions of several spatial variables, as well as of time. The transformations $Tr(F_{i})$ have two goals: (1) They can remove some non-linearities if possible (an example is the Hopf–Cole transformation, which leads to the linearization of the Burger’s equation); (2) They can transform the non-linearity of the solved differential equations to a more treatable kind of non-linearity (e.g., to polynomial non-linearity). In many particular cases one may skip this step (then we have just u(x,…,t)=F(x,…,t)), but in numerous cases this step is necessary for obtaining a solution of the studied NPDE. The substitution of Equation (Equation 3) in Equation (Equation 2) leads to a non-linear PDE for the function F(x,…,t). In many cases, the general form of the transformation Tr(F) is not known.

(2). This step is based on the use of composite functions. In this step, the functions $F_{1}\left(x,\ldots ,t\right),F_{2}\left(x,\ldots ,t\right),\ldots$ are chosen as composite functions of the functions $f_{i1},\ldots ,f_{iN},\ldots$, which are solutions of simpler differential equations. There are two possibilities: (1) The construction relationship for the composite function is not fixed. Then, the Faa di Bruno relationship for the derivatives of a composite function is used; (2) The construction relationship for the composite function is fixed. For example, for the case of one solved equation and one function *F*, the construction relationship can be given as:(4)$$F=\hat{\alpha }+\sum_{i_{1}=1}^{N}{\hat{\beta }}_{i}f_{i_{1}}+\sum_{i_{1}=1}^{N}\sum_{i_{2}=1}^{N}{\hat{\gamma }}_{i_{1}}f_{i_{1}}f_{i_{2}}+\ldots +\sum_{i_{1}=1}^{N}\ldots \sum_{i_{N}=1}^{N}{\hat{\sigma }}_{i_{1}\ldots_{n}}f_{i_{1}}\ldots f_{i_{N}}$$Then, one can directly calculate the corresponding derivatives from the solved differential equation.

(3). In this step, the simple equations for the functions $f_{i_{1}},\ldots ,f_{i_{N}}$ must be selected. In addition, in accordance with the hypothesis of Point (1) of Step 2, the relationship between the composite functions $F_{1}\left(x,\ldots ,t\right),\ldots ,F_{N}\left(x,\ldots ,t\right)$ and the functions $f_{i_{1}},\ldots ,f_{i_{N}}$ must be fixed. The fixation of the simple equations and the fixation of the relationships for the composite functions are connected. The fixations transform the left-hand sides of Equation (Equation 2). The result of this transformation can be functions that are the sum of terms. Each of these terms contains some function multiplied by a coefficient. This coefficient is a relationship containing some of the parameters of the solved equations and some of the parameters of the solutions of the simple equations used. The fixation mentioned above is performed by a balance procedure that ensures that the relationships for the coefficients contain more than one term. This balance procedure leads to one or more additional relationships among the parameters of the solved equation and parameters of the solutions of the simple equations used. These additional relationships are known as balance equations.

(4). A non-trivial solution of Equation (Equation 2) is obtained if all coefficients mentioned in Step 3 are set to zero. This condition usually leads to a system of non-linear algebraic equations. The unknown variables in these equations are the coefficients of the solved non-linear differential equation and the coefficients of the solutions of the simple equations. Any non-trivial solution of this algebraic system leads to a solution of the studied non-linear PDE.

Below, we shall apply the methodology above given to obtain exact solutions of Equation (Equation 1). We shall consider *u* as a composite function of two functions of two variables, i.e.,
(5)$$u\left(\xi_{1},\xi_{2}\right)=1+F_{1}\left(\xi_{1}\right)+F_{2}\left(\xi_{2}\right),$$
where
(6)$$\xi_{1}=\kappa_{1}x+\omega_{1}t,\;\;\xi_{2}=\kappa_{2}x+\omega_{2}t,$$
as
(7)$$F_{1}\left(\xi_{1}\right)=\sum_{i_{1}=0}^{n_{1}}\zeta_{i_{1}}{\left[f_{1}\left(\xi_{1}\right)\right]}^{i_{1}},\;F_{2}\left(\xi_{2}\right)=\sum_{i_{2}=0}^{n_{2}}\eta_{i_{2}}{\left[f_{2}\left(\xi_{2}\right)\right]}^{i_{2}}$$
where $\zeta_{i_{1}},\;i_{1}=0,\ldots ,n_{1}$ and $\eta_{i_{2}},\;i_{2}=0,\ldots ,n_{2}$ are parameters, and $n_{1}$ and $n_{2}$ shall be determined by means of balance procedure. Let us present the solutions of functions $f_{1}$ and $f_{2}$ by the special functions $V_{\mu_{0},\mu_{1},\ldots ,\mu_{m_{1}}}\left(\xi_{1};k_{1},l_{1},m_{1}\right)$ and $V_{\nu_{0},\nu_{1},\ldots ,\nu_{m_{2}}}\left(\xi_{2};k_{2},l_{2},m_{2}\right)$, which are solutions of the simple equations of the following kind:(8)$${\left(\frac{d^{k_{1}}f_{1}}{d\xi^{k_{1}}}\right)}^{l_{1}}=\sum_{j_{1}=0}^{m_{1}}\mu_{j_{1}}f_{1}^{j_{1}},\;\;\;{\left(\frac{d^{k_{2}}f_{2}}{d\xi^{k_{2}}}\right)}^{l_{2}}=\sum_{j_{2}=0}^{m_{2}}\nu_{j_{2}}f_{2}^{j_{2}}$$
where $k_{1,2}$ are the orders of derivatives of $f_{1}$ and $f_{2}$, $l_{1,2}$ are the degrees of derivatives in the defining ODEs and $m_{1,2}$ are the highest degrees of the polynomials of $f_{1}$ and $f_{2}$ in the defining ODE. The special functions $V_{\mu_{0},\mu_{1},\ldots ,\mu_{m_{1}}}\left(\xi_{1};k_{1},l_{1},m_{1}\right)$ and $V_{\nu_{0},\nu_{1},\ldots ,\nu_{m_{2}}}\left(\xi_{2};k_{2},l_{2},m_{2}\right)$ have interesting properties. These functions can be hyperbolic, trigonometric, elliptic functions of Jacobi, etc. For our study, we choose one specific case of the functions *V*. We shall assume that $k_{1}=k_{2}=1$ and $l_{1}=l_{2}=1$. Then, the functions $V_{\mu_{0},\mu_{1},\ldots ,\mu_{m_{1}}}\left(\xi_{1};1,1,m_{1}\right)$ and $V_{\nu_{0},\nu_{1},\ldots ,\nu_{m_{2}}}\left(\xi_{2};1,1,m_{2}\right)$ are solutions of the simple equations: (9)$$\frac{df_{1}}{d\xi_{1}}=\sum_{j_{1}=0}^{m_{1}}\mu_{j_{1}}f_{1}^{j_{1}},\;\;\;\frac{df_{2}}{d\xi_{2}}=\sum_{j_{2}=0}^{m_{2}}\nu_{j_{2}}f_{2}^{j_{2}}$$

In the study, we shall present various examples of application of the special functions *V* depending on the numerical value of $m_{1}$ and $m_{2}$. We shall use the following general types of simple equations:The Bernoulli equation, whose general form is:
(10)$$\frac{df}{d\xi }=af\left(\xi \right)+b{\left[f\left(\xi \right)\right]}^{m}$$The general solution of this equation is:
(11)$$f\left(\xi \right)={\left(\frac{a\exp[a\left(m-1\right)\left(\xi +\xi_{0}\right)]}{1-b\exp[a\left(m-1\right)\left(\xi +\xi_{0}\right)]}\right)}^{\frac{1}{m-1}}$$
for the case a>0,b<0 and
(12)$$f\left(\xi \right)={\left(\frac{a\exp[a\left(m-1\right)\left(\xi +\xi_{0}\right)]}{1+b\exp[a\left(m-1\right)\left(\xi +\xi_{0}\right)]}\right)}^{\frac{1}{m-1}}$$
for the case a<0,b>0, as $\xi_{0}$ is a constant of integration.The Abel equation of first kind, whose general form is:
(13)$$\frac{df}{d\xi }=a+bf\left(\xi \right)+c{\left[f\left(\xi \right)\right]}^{2}+d{\left[f\left(\xi \right)\right]}^{3}$$For the special case $a=\frac{c}{3d}\left(b-\frac{2c^{2}}{9d}\right)$, this equation has the following solution:
(14)$$f\left(\xi \right)=\frac{\exp\left[\left(b-\frac{c^{2}}{3d}\right)\left(\xi +\xi_{0}\right)\right]}{\sqrt{C^{*}-d\exp\left[2\left(b-\frac{c^{2}}{3d}\right)\left(\xi +\xi_{0}\right)\right]}}-\frac{c}{3d}$$
where $C^{*}$ and $\xi_{0}$ are constants of integration.The Riccati equation, whose general form is:
(15)$$\frac{df}{d\xi }=a{\left[f\left(\xi \right)\right]}^{2}+bf\left(\xi \right)+c$$The general solutions of this equation are:
(16)$$f\left(\xi \right)=-\frac{b}{2a}-\frac{\theta }{2a}\tanh\left[\frac{\theta (\xi +\xi_{0})}{2}\right]$$
and
(17)$$\begin{array}{c} f\left(\xi \right)=-\frac{b}{2a}-\frac{\theta }{2a}\tanh\left[\frac{\theta (\xi +\xi_{0})}{2}\right]\\ +\frac{\exp[\frac{\theta (\xi +\xi_{0})}{2}]}{2\cosh\left[\frac{\theta (\xi +\xi_{0})}{2}\right]\left[\frac{a}{\theta }+2C^{*}\exp\left[\frac{\theta (\xi +\xi_{0})}{2}\right]\cosh\left[\frac{\theta (\xi +\xi_{0})}{2}\right]\right]} \end{array}$$
where $\theta^{2}=b^{2}-4ac>0$ and $C^{*}$ and $\xi_{0}$ are constants of integration. In this study, we shall use only the extended variant of the Riccati equation (Equation (Equation 17)). In addition, as a particular case of the use of equations of Riccati as simple equations, we shall consider also the so-called extended tanh-function equation:
(18)$$\frac{df}{d\xi }={\bar{c}}^{2}-{\bar{a}}^{2}{\left[f\left(\xi \right)\right]}^{2}$$Equation (Equation 18) is obtained from Equation (Equation 15) when $b=0,\;a=-{\bar{a}}^{2},\;c={\bar{c}}^{2}$ and its solution is:
(19)$$f\left(\xi \right)=\frac{\bar{c}}{\bar{a}}\tanh\left[\bar{a}\bar{c}\left(\xi +\xi_{0}\right)\right],$$
where ${\bar{a}}^{2}f{\left(\xi \right)}^{2}<{\bar{c}}^{2}$ and $\xi_{0}$ is a constant of integration.The linear ODE, which has the following form:
(20)$$\frac{df}{d\xi }=af\left(\xi \right)+b,$$
and its solution is:
(21)$$f\left(\xi \right)=C^{*}\exp\left[a\left(\xi +\xi_{0}\right)\right]-\frac{b}{a},$$
where $C^{*}$ and $\xi_{0}$ are constants.

## 3. Exact Solutions of the Extended KdV Equation

Following the above given algorithm, we skip Step 1 of the SEsM (no additional transformation of non-linearity). In Step 2, we consider *u* as a composite function of two functions of two variables (see Equation (Equation 5)). Substitution of Equations (Equation 5)–(Equation 7) in Equation (Equation 1) leads to the following ODE:(22)$$\begin{array}{c} \left(\omega_{1}+\omega_{2}\right)\left(\frac{dF_{1}}{d\xi_{1}}+\frac{dF_{2}}{d\xi_{2}}\right)+\left({\kappa_{1}}^{5}+{\kappa_{2}}^{5}\right)\left(\frac{d^{5}F_{1}}{d\xi_{1}^{5}}+\frac{d^{5}F_{2}}{d\xi_{2}^{5}}\right)\\ -\alpha \;\left({\kappa_{1}}^{4}+{\kappa_{2}}^{4}\right)\left(1+F_{1}+F_{2}\right)\;\left(\frac{d^{3}F_{1}}{d\xi_{1}^{3}}+\frac{d^{3}F_{2}}{d\xi_{2}^{3}}\right)\\ +\beta \;\left({\kappa_{1}}^{3}+{\kappa_{2}}^{3}\right)\left(\frac{dF_{1}}{d\xi_{1}}+\frac{dF_{2}}{d\xi_{2}}\right)\;\left(\frac{d^{2}F_{1}}{d\xi_{1}^{2}}+\frac{d^{2}F_{2}}{d\xi_{2}^{2}}\right)\\ +\gamma \;\left(\kappa_{1}+\kappa_{2}\right){\left(1+F_{1}+F_{2}\right)}^{2}\left(\frac{dF_{1}}{d\xi_{1}}+\frac{dF_{2}}{d\xi_{2}}\right)+\delta \;\left({\kappa_{1}}^{3}+{\kappa_{2}}^{3}\right)\left(\frac{d^{3}F_{1}}{d\xi_{1}^{3}}+\frac{d^{3}F_{2}}{d\xi_{2}^{3}}\right)\\ +\epsilon \;\left(\kappa_{1}+\kappa_{2}\right)\left(1+F_{1}+F_{2}\right)\;\left(\frac{dF_{1}}{d\xi_{1}}+\frac{dF_{2}}{d\xi_{2}}\right)=0 \end{array}$$In Step 3 of the SEsM, we must select the equation for $u(F_{1}\left[f_{1}\left(\xi_{1}\right)\right],F_{2}\left[f_{2}\left(\xi_{2}\right)\right])$ (the relationship for the composite function) and the equations for $f_{1}^{\prime }\left(\xi_{1}\right)$ and $f_{1}^{\prime }\left(\xi_{2}\right)$ (the simple equations). We assume that the expression for *u* is of kind (Equation 5). In addition, the simple equations are assumed to be of kind (Equation 9). The substitution of Equations (Equation 5), (Equation 7) and (Equation 9) in Equation (Equation 22) leads to polynomials of the functions $f_{1}$ and $f_{2}$. To obtain the system of non-linear algebraic equations, we must balance the largest degrees of these polynomials. This procedure leads to the balance equations
(23)$$n_{1}=2m_{1}-2,\;\;n_{2}=2m_{2}-2,$$

Then Equation (Equation 1) may have solutions of the kind
(24)$$u\left(\xi_{1},\xi_{2}\right)=1+\sum_{i_{1}=0}^{2m_{1}-2}\zeta_{i_{1}}{\left[f_{1}\left(\xi_{1}\right)\right]}^{i_{1}}+\sum_{i_{2}=0}^{2m_{2}-2}\eta_{i_{2}}{\left[f_{2}\left(\xi_{2}\right)\right]}^{i_{2}}$$
and the functions $f_{1}\left(\xi_{1}\right)$ and $f_{2}\left(\xi_{2}\right)$ are solutions of the simple Equation (Equation 9).

### 3.1. Case $m_{1}=3,\;m_{2}=3$

First, we shall search for analytical solutions of Equation (Equation 1) for $m_{1}=3,\;m_{2}=3$. According to the balance Equation (Equation 23), $n_{1}=4,\;n_{2}=4$. The general solution of Equation (Equation 1) can be written as
(25)$$u\left(\xi_{1},\xi_{2}\right)=1+\sum_{i_{1}=0}^{4}\zeta_{i_{1}}{\left[f_{1}\left(\xi_{1}\right)\right]}^{i_{1}}+\sum_{i_{2}=0}^{4}\eta_{i_{2}}{\left[f_{2}\left(\xi_{2}\right)\right]}^{i_{2}}$$
where
(26)$$\frac{df_{1}}{d\xi_{1}}=\mu_{0}+\mu_{1}f_{1}+\mu_{2}f_{1}^{2}+\mu_{3}f_{1}^{3},\;\;\frac{df_{2}}{d\xi_{2}}=\nu_{0}+\nu_{1}f_{2}+\nu_{2}f_{2}^{2}+\nu_{3}f_{2}^{3}$$Substitution of Equations (Equation 25) and (Equation 26) in Equation (Equation 22) leads to the following system of non-linear algebraic equations: (27)$$\begin{array}{c} 96\;\beta \;{\kappa_{1}}^{3}{\zeta_{4}}^{2}{\mu_{3}}^{3}+23040\;{\kappa_{2}}^{5}\zeta_{4}{\mu_{3}}^{5}+96\;\beta \;{\kappa_{2}}^{3}{\zeta_{4}}^{2}{\mu_{3}}^{3}+4\;\gamma \;\kappa_{1}{\zeta_{4}}^{3}\mu_{3}\\ -192\;\alpha \;{\kappa_{2}}^{4}{\zeta_{4}}^{2}{\mu_{3}}^{3}+4\;\sigma \;\kappa_{2}{\zeta_{4}}^{3}\mu_{3}+23040\;{\kappa_{1}}^{5}\zeta_{4}{\mu_{3}}^{5}-192\;\alpha \;{\kappa_{1}}^{4}{\zeta_{4}}^{2}{\mu_{3}}^{3}=0 \end{array}$$
$$\begin{array}{c} -500\;\alpha \;{\kappa_{2}}^{4}{\zeta_{4}}^{2}{\mu_{3}}^{2}\mu_{2}-500\;\alpha \;{\kappa_{1}}^{4}{\zeta_{4}}^{2}{\mu_{3}}^{2}\mu_{2}-297\;\alpha \;{\kappa_{1}}^{4}\zeta_{3}\zeta_{4}{\mu_{3}}^{3}+11\;\gamma \;\kappa_{2}\zeta_{3}{\zeta_{4}}^{2}\mu_{3}\\ +272\;\beta \;{\kappa_{1}}^{3}{\zeta_{4}}^{2}{\mu_{3}}^{2}\mu_{2}+4\;\gamma \;\kappa_{2}{\zeta_{4}}^{3}\mu_{2}+11\;\gamma \;\kappa_{1}\zeta_{3}{\zeta_{4}}^{2}\mu_{3}+132\;\beta \;{\kappa_{1}}^{3}\zeta_{4}{\mu_{3}}^{3}\zeta_{3}\\ +93660\;{\kappa_{1}}^{5}\zeta_{4}{\mu_{3}}^{4}\mu_{2}+10395\;{\kappa_{1}}^{5}\zeta_{3}{\mu_{3}}^{5}-297\;\alpha \;{\kappa_{2}}^{4}\zeta_{3}\zeta_{4}{\mu_{3}}^{3}+10395\;{\kappa_{2}}^{5}\zeta_{3}{\mu_{3}}^{5}\\ +132\;\beta \;{\kappa_{2}}^{3}\zeta_{4}{\mu_{3}}^{3}\zeta_{3}+93660\;{\kappa_{2}}^{5}\zeta_{4}{\mu_{3}}^{4}\mu_{2}+272\;\beta \;{\kappa_{2}}^{3}{\zeta_{4}}^{2}{\mu_{3}}^{2}\mu_{2}+4\;\gamma \;\kappa_{1}{\zeta_{4}}^{3}\mu_{2}=0 \end{array}$$
$$\begin{array}{c} -105\;\alpha \;{\kappa_{2}}^{4}{\zeta_{3}}^{2}{\mu_{3}}^{3}+45\;\beta \;{\kappa_{1}}^{3}{\zeta_{3}}^{2}{\mu_{3}}^{3}+45\;\beta \;{\kappa_{2}}^{3}{\zeta_{3}}^{2}{\mu_{3}}^{3}+41205\;{\kappa_{1}}^{5}{\mu_{3}}^{4}\zeta_{3}\mu_{2}+76800\;{\kappa_{1}}^{5}\zeta_{4}{\mu_{3}}^{4}\mu_{1}\\ +149860\;{\kappa_{1}}^{5}{\mu_{3}}^{3}\zeta_{4}{\mu_{2}}^{2}+41205\;{\kappa_{2}}^{5}{\mu_{3}}^{4}\zeta_{3}\mu_{2}+76800\;{\kappa_{2}}^{5}\zeta_{4}{\mu_{3}}^{4}\mu_{1}=0 \end{array}$$
$$\begin{array}{c} 149860\;{\kappa_{2}}^{5}{\mu_{3}}^{3}\zeta_{4}{\mu_{2}}^{2}-105\;\alpha \;{\kappa_{1}}^{4}{\zeta_{3}}^{2}{\mu_{3}}^{3}+4\;\gamma \;\kappa_{1}{\zeta_{4}}^{3}\mu_{1}+3840\;{\kappa_{1}}^{5}\zeta_{2}{\mu_{3}}^{5}\\ +3840\;{\kappa_{2}}^{5}\zeta_{2}{\mu_{3}}^{5}+10\;\gamma \;\kappa_{2}\zeta_{2}{\zeta_{4}}^{2}\mu_{3}+10\;\gamma \;\kappa_{2}{\zeta_{3}}^{2}\zeta_{4}\mu_{3}+11\;\gamma \;\kappa_{2}\zeta_{3}{\zeta_{4}}^{2}\mu_{2}=0 \end{array}$$
$$\begin{array}{c} 10\;\gamma \;\kappa_{1}\zeta_{2}{\zeta_{4}}^{2}\mu_{3}+10\;\gamma \;\kappa_{1}{\zeta_{3}}^{2}\zeta_{4}\mu_{3}+11\;\gamma \;\kappa_{1}\zeta_{3}{\zeta_{4}}^{2}\mu_{2}-240\;\alpha \;{\kappa_{1}}^{4}\zeta_{2}\zeta_{4}{\mu_{3}}^{3}-767\;\alpha \;{\kappa_{1}}^{4}\zeta_{3}\zeta_{4}{\mu_{3}}^{2}\mu_{2}\\ -240\;\alpha \;{\kappa_{2}}^{4}\zeta_{2}\zeta_{4}{\mu_{3}}^{3}-432\;\alpha \;{\kappa_{2}}^{4}{\zeta_{4}}^{2}{\mu_{3}}^{2}\mu_{1}-428\;\alpha \;{\kappa_{2}}^{4}{\zeta_{4}}^{2}\mu_{3}{\mu_{2}}^{2}=0 \end{array}$$
$$\begin{array}{c} -432\;\alpha \;{\kappa_{1}}^{4}{\zeta_{4}}^{2}{\mu_{3}}^{2}\mu_{1}-428\;\alpha \;{\kappa_{1}}^{4}{\zeta_{4}}^{2}\mu_{3}{\mu_{2}}^{2}+80\;\beta \;\kappa_{1}^{3}\;\zeta_{4}\;\mu_{3}^{3}\;\zeta_{2}+256\;\beta \;\kappa_{1}^{3}\;\zeta_{4}^{2}\;\mu_{3}\;\mu_{2}^{2}\\ +256\;\beta \;{\kappa_{1}}^{3}{\zeta_{4}}^{2}{\mu_{3}}^{2}\mu_{1}+80\;\beta \;{\kappa_{2}}^{3}\zeta_{4}{\mu_{3}}^{3}\zeta_{2}+256\;\beta \;{\kappa_{2}}^{3}{\zeta_{4}}^{2}\mu_{3}{\mu_{2}}^{2}\\ +256\;\beta \;{\kappa_{2}}^{3}{\zeta_{4}}^{2}{\mu_{3}}^{2}\mu_{1}-767\;\alpha \;{\kappa_{2}}^{4}\zeta_{3}\zeta_{4}{\mu_{3}}^{2}\mu_{2}=0 \end{array}$$
$$\begin{array}{c} 372\;\beta \;{\kappa_{2}}^{3}\zeta_{4}{\mu_{3}}^{2}\zeta_{3}\mu_{2}+372\;\beta \;{\kappa_{1}}^{3}\zeta_{4}{\mu_{3}}^{2}\zeta_{3}\mu_{2}+4\;\gamma \;\kappa_{1}{\zeta_{4}}^{3}\mu_{1}+4\;\gamma \;\kappa_{2}{\zeta_{4}}^{3}\mu_{1}=0 \end{array}$$
$$\begin{array}{c} 11\;\gamma \;\kappa_{1}\zeta_{3}{\zeta_{4}}^{2}\mu_{1}+945\;{\kappa_{1}}^{5}\zeta_{1}{\mu_{3}}^{5}+945\;{\kappa_{2}}^{5}\zeta_{1}{\mu_{3}}^{5}+9\;\gamma \;\kappa_{2}\zeta_{1}{\zeta_{4}}^{2}\mu_{3}\\ +10\;\gamma \;\kappa_{2}\zeta_{2}{\zeta_{4}}^{2}\mu_{2}+10\;\gamma \;\kappa_{2}{\zeta_{3}}^{2}\zeta_{4}\mu_{2}-618\;\alpha \;{\kappa_{1}}^{4}\zeta_{2}\zeta_{4}{\mu_{3}}^{2}\mu_{2}-657\;\alpha \;{\kappa_{1}}^{4}\zeta_{3}\zeta_{4}{\mu_{3}}^{2}\mu_{1}\\ -650\;\alpha \;{\kappa_{1}}^{4}\zeta_{3}\mu_{3}\zeta_{4}{\mu_{2}}^{2}-728\;\alpha \;{\kappa_{1}}^{4}{\zeta_{4}}^{2}\mu_{3}\mu_{2}\mu_{1}+241848\;{\kappa_{1}}^{5}{\mu_{3}}^{3}\mu_{2}\zeta_{4}\mu_{1}\\ +241848\;{\kappa_{2}}^{5}{\mu_{3}}^{3}\mu_{2}\zeta_{4}\mu_{1}-207\;\alpha \;{\kappa_{1}}^{4}\zeta_{1}\zeta_{4}{\mu_{3}}^{3}-207\;\alpha \;{\kappa_{2}}^{4}\zeta_{1}\zeta_{4}{\mu_{3}}^{3}-120\;\alpha \;{\kappa_{2}}^{4}{\zeta_{4}}^{2}{\mu_{2}}^{3}=0 \end{array}$$
$$\begin{array}{c} 80\;\beta \;{\kappa_{1}}^{3}{\zeta_{4}}^{2}{\mu_{2}}^{3}+80\;\beta \;{\kappa_{2}}^{3}{\zeta_{4}}^{2}{\mu_{2}}^{3}-267\;\alpha \;{\kappa_{1}}^{4}{\zeta_{3}}^{2}{\mu_{3}}^{2}\mu_{2}-372\;\alpha \;{\kappa_{1}}^{4}{\zeta_{4}}^{2}{\mu_{3}}^{2}\mu_{0}\\ -153\;\alpha \;{\kappa_{2}}^{4}\zeta_{2}\zeta_{3}{\mu_{3}}^{3}-153\;\alpha \;{\kappa_{1}}^{4}\zeta_{2}\zeta_{3}{\mu_{3}}^{3}-267\;\alpha \;{\kappa_{2}}^{4}{\zeta_{3}}^{2}{\mu_{3}}^{2}\mu_{2}-372\;\alpha \;{\kappa_{2}}^{4}{\zeta_{4}}^{2}{\mu_{3}}^{2}\mu_{0}\\ +36\;\beta \;{\kappa_{1}}^{3}\zeta_{4}{\mu_{3}}^{3}\zeta_{1}+240\;\beta \;{\kappa_{1}}^{3}{\zeta_{4}}^{2}{\mu_{3}}^{2}\mu_{0}+54\;\beta \;{\kappa_{1}}^{3}\zeta_{3}{\mu_{3}}^{3}\zeta_{2}=0 \end{array}$$
$$\begin{array}{c} 126\;\beta \;{\kappa_{1}}^{3}{\zeta_{3}}^{2}{\mu_{3}}^{2}\mu_{2}+36\;\beta \;{\kappa_{2}}^{3}\zeta_{4}{\mu_{3}}^{3}\zeta_{1}+240\;\beta \;{\kappa_{2}}^{3}{\zeta_{4}}^{2}{\mu_{3}}^{2}\mu_{0}+54\;\beta \;{\kappa_{2}}^{3}\zeta_{3}{\mu_{3}}^{3}\zeta_{2}\\ +126\;\beta \;{\kappa_{2}}^{3}{\zeta_{3}}^{2}{\mu_{3}}^{2}\mu_{2}-728\;\alpha \;{\kappa_{2}}^{4}{\zeta_{4}}^{2}\mu_{3}\mu_{2}\mu_{1}-618\;\alpha \;{\kappa_{2}}^{4}\zeta_{2}\zeta_{4}{\mu_{3}}^{2}\mu_{2}-657\;\alpha \;{\kappa_{2}}^{4}\zeta_{3}\zeta_{4}{\mu_{3}}^{2}\mu_{1}\\ -650\;\alpha \;{\kappa_{2}}^{4}\zeta_{3}\mu_{3}\zeta_{4}{\mu_{2}}^{2}+4\;\gamma \;\kappa_{1}{\zeta_{4}}^{3}\mu_{0}+4\;\gamma \;\kappa_{2}{\zeta_{4}}^{3}\mu_{0}+3\;\gamma \;\kappa_{2}{\zeta_{3}}^{3}\mu_{3}=0 \end{array}$$
$$\begin{array}{c} 224\;\beta \;{\kappa_{2}}^{3}\zeta_{4}{\mu_{3}}^{2}\zeta_{2}\mu_{2}+348\;\beta \;{\kappa_{2}}^{3}\zeta_{4}{\mu_{3}}^{2}\zeta_{3}\mu_{1}+480\;\beta \;{\kappa_{2}}^{3}{\zeta_{4}}^{2}\mu_{3}\mu_{2}\mu_{1}+348\;\beta \;{\kappa_{2}}^{3}\zeta_{4}\mu_{3}\zeta_{3}{\mu_{2}}^{2}\\ +224\;\beta \;{\kappa_{1}}^{3}\zeta_{4}{\mu_{3}}^{2}\zeta_{2}\mu_{2}+348\;\beta \;{\kappa_{1}}^{3}\zeta_{4}\mu_{3}\zeta_{3}{\mu_{2}}^{2}+348\;\beta \;{\kappa_{1}}^{3}\zeta_{4}{\mu_{3}}^{2}\zeta_{3}\mu_{1}+480\;\beta \;{\kappa_{1}}^{3}{\zeta_{4}}^{2}\mu_{3}\mu_{2}\mu_{1}\\ +11\;\gamma \;\kappa_{2}\zeta_{3}{\zeta_{4}}^{2}\mu_{1}+9\;\gamma \;\kappa_{1}\zeta_{1}{\zeta_{4}}^{2}\mu_{3}+10\;\gamma \;\kappa_{1}\zeta_{2}{\zeta_{4}}^{2}\mu_{2}+10\;\gamma \;\kappa_{1}{\zeta_{3}}^{2}\zeta_{4}\mu_{2}=0 \end{array}$$
$$\begin{array}{c} 14730\;{\kappa_{1}}^{5}{\mu_{3}}^{4}\zeta_{2}\mu_{2}+33075\;{\kappa_{1}}^{5}{\mu_{3}}^{4}\zeta_{3}\mu_{1}+63756\;{\kappa_{1}}^{5}\zeta_{4}{\mu_{3}}^{4}\mu_{0}+64020\;{\kappa_{1}}^{5}{\mu_{3}}^{3}\zeta_{3}{\mu_{2}}^{2}\\ +117616\;{\kappa_{1}}^{5}{\mu_{3}}^{2}\zeta_{4}{\mu_{2}}^{3}+14730\;{\kappa_{2}}^{5}{\mu_{3}}^{4}\zeta_{2}\mu_{2}+33075\;{\kappa_{2}}^{5}{\mu_{3}}^{4}\zeta_{3}\mu_{1}+63756\;{\kappa_{2}}^{5}\zeta_{4}{\mu_{3}}^{4}\mu_{0}\\ +64020\;{\kappa_{2}}^{5}{\mu_{3}}^{3}\zeta_{3}{\mu_{2}}^{2}-120\;\alpha \;{\kappa_{1}}^{4}{\zeta_{4}}^{2}{\mu_{2}}^{3}+18\;\gamma \;\kappa_{1}\zeta_{2}\zeta_{3}\zeta_{4}\mu_{3}\\ -117616\;{\kappa_{2}}^{5}{\mu_{3}}^{2}\zeta_{4}{\mu_{2}}^{3}\gamma \;\kappa_{1}{\zeta_{3}}^{3}\mu_{3}+18\;\gamma \;\kappa_{2}\zeta_{2}\zeta_{3}\zeta_{4}\mu_{3}=0 \end{array}$$
$$\begin{array}{c} 8\;\gamma \;\kappa_{2}{\zeta_{4}}^{2}\eta_{4}\mu_{3}-192\;\alpha \;{\kappa_{2}}^{4}\eta_{4}\zeta_{4}{\mu_{3}}^{3}-192\;\alpha \;{\kappa_{1}}^{4}\eta_{4}\zeta_{4}{\mu_{3}}^{3}+8\;\gamma \;\kappa_{1}{\zeta_{4}}^{2}\eta_{4}\mu_{3}=0\\ 8\;\gamma \;\kappa_{1}{\zeta_{4}}^{2}\eta_{1}\mu_{3}+8\;\gamma \;\kappa_{2}{\zeta_{4}}^{2}\eta_{1}\mu_{3}-192\;\alpha \;{\kappa_{2}}^{4}\eta_{1}\zeta_{4}{\mu_{3}}^{3}-192\;\alpha \;{\kappa_{1}}^{4}\eta_{1}\zeta_{4}{\mu_{3}}^{3}=0 \end{array}$$
$$\begin{array}{c} 8\;\gamma \;\kappa_{1}\zeta_{2}{\zeta_{3}}^{2}\mu_{3}+9\;\gamma \;\kappa_{2}\zeta_{1}{\zeta_{4}}^{2}\mu_{2}+8\;\gamma \;\kappa_{2}\zeta_{2}{\zeta_{3}}^{2}\mu_{3}+8\;\gamma \;\kappa_{2}{\zeta_{2}}^{2}\zeta_{4}\mu_{3}\\ +9\;\gamma \;\kappa_{1}\zeta_{1}{\zeta_{4}}^{2}\mu_{2}+8\;\gamma \;\kappa_{2}\zeta_{0}{\zeta_{4}}^{2}\mu_{3}+8\;\gamma \;\kappa_{1}\zeta_{0}{\zeta_{4}}^{2}\mu_{3}+8\;\gamma \;\kappa_{1}{\zeta_{2}}^{2}\zeta_{4}\mu_{3}\\ +10\;\gamma \;\kappa_{1}\zeta_{2}{\zeta_{4}}^{2}\mu_{1}+10\;\gamma \;\kappa_{1}{\zeta_{3}}^{2}\zeta_{4}\mu_{1}+11\;\gamma \;\kappa_{1}\zeta_{3}{\zeta_{4}}^{2}\mu_{0}+8\;\gamma \;\kappa_{1}{\zeta_{4}}^{2}\eta_{0}\mu_{3}\\ +10\;\gamma \;\kappa_{2}\zeta_{2}{\zeta_{4}}^{2}\mu_{1}+10\;\gamma \;\kappa_{2}{\zeta_{3}}^{2}\zeta_{4}\mu_{1}+11\;\gamma \;\kappa_{2}\zeta_{3}{\zeta_{4}}^{2}\mu_{0}+8\;\gamma \;\kappa_{2}{\zeta_{4}}^{2}\eta_{0}\mu_{3}=0 \end{array}$$
$$\begin{array}{c} 18\;\gamma \;\kappa_{2}\zeta_{2}\zeta_{3}\zeta_{4}\mu_{2}+18\;\gamma \;\kappa_{1}\zeta_{2}\zeta_{3}\zeta_{4}\mu_{2}+16\;\gamma \;\kappa_{1}\zeta_{1}\zeta_{3}\zeta_{4}\mu_{3}+16\;\gamma \;\kappa_{2}\zeta_{1}\zeta_{3}\zeta_{4}\mu_{3}\\ +45096\;{\kappa_{2}}^{5}\mu_{3}\zeta_{4}{\mu_{2}}^{4}-192\;\alpha \;{\kappa_{1}}^{4}\zeta_{4}{\mu_{3}}^{3}-48\;\alpha \;{\kappa_{1}}^{4}{\zeta_{2}}^{2}{\mu_{3}}^{3}-192\;\alpha \;{\kappa_{2}}^{4}\zeta_{4}{\mu_{3}}^{3}\\ +3465\;{\kappa_{1}}^{5}{\mu_{3}}^{4}\zeta_{1}\mu_{2}+11520\;{\kappa_{1}}^{5}{\mu_{3}}^{4}\zeta_{2}\mu_{1}+27027\;{\kappa_{1}}^{5}{\mu_{3}}^{4}\zeta_{3}\mu_{0}+22010\;{\kappa_{1}}^{5}{\mu_{3}}^{3}\zeta_{2}{\mu_{2}}^{2}\\ +96000\;{\kappa_{1}}^{5}{\mu_{3}}^{3}\zeta_{4}{\mu_{1}}^{2}+48522\;{\kappa_{1}}^{5}{\mu_{3}}^{2}\zeta_{3}{\mu_{2}}^{3}+45096\;{\kappa_{1}}^{5}\mu_{3}\zeta_{4}{\mu_{2}}^{4}+3465\;{\kappa_{2}}^{5}{\mu_{3}}^{4}\zeta_{1}\mu_{2}\\ +11520\;{\kappa_{2}}^{5}{\mu_{3}}^{4}\zeta_{2}\mu_{1}+27027\;{\kappa_{2}}^{5}{\mu_{3}}^{4}\zeta_{3}\mu_{0}+22010\;{\kappa_{2}}^{5}{\mu_{3}}^{3}\zeta_{2}{\mu_{2}}^{2}\\ +96000\;{\kappa_{2}}^{5}{\mu_{3}}^{3}\zeta_{4}{\mu_{1}}^{2}+48522\;{\kappa_{2}}^{5}{\mu_{3}}^{2}\zeta_{3}{\mu_{2}}^{3}=0 \end{array}$$
$$\begin{array}{c} 14\;\gamma \;\kappa_{2}\zeta_{3}\eta_{4}\zeta_{4}\mu_{3}-500\;\alpha \;{\kappa_{2}}^{4}\eta_{4}\zeta_{4}{\mu_{3}}^{2}\mu_{2}-105\;\alpha \;{\kappa_{1}}^{4}\eta_{4}\zeta_{3}{\mu_{3}}^{3}+8\;\gamma \;\kappa_{1}{\zeta_{4}}^{2}\eta_{4}\mu_{2}\\ -500\;\alpha \;{\kappa_{1}}^{4}\eta_{4}\zeta_{4}{\mu_{3}}^{2}\mu_{2}+14\;\gamma \;\kappa_{1}\zeta_{3}\eta_{4}\zeta_{4}\mu_{3}+8\;\gamma \;\kappa_{2}{\zeta_{4}}^{2}\eta_{4}\mu_{2}-105\;\alpha \;{\kappa_{2}}^{4}\eta_{4}\zeta_{3}{\mu_{3}}^{3}=0 \end{array}$$
$$\begin{array}{c} 8\;\gamma \;\kappa_{1}{\zeta_{4}}^{2}\eta_{3}\mu_{2}-500\;\alpha \;{\kappa_{1}}^{4}\eta_{3}\zeta_{4}{\mu_{3}}^{2}\mu_{2}+14\;\gamma \;\kappa_{2}\zeta_{3}\eta_{3}\zeta_{4}\mu_{3}-500\;\alpha \;{\kappa_{2}}^{4}\eta_{3}\zeta_{4}{\mu_{3}}^{2}\mu_{2}\\ -105\;\alpha \;{\kappa_{2}}^{4}\eta_{3}\zeta_{3}{\mu_{3}}^{3}+14\;\gamma \;\kappa_{1}\zeta_{3}\eta_{3}\zeta_{4}\mu_{3}+8\;\gamma \;\kappa_{2}{\zeta_{4}}^{2}\eta_{3}\mu_{2}-105\;\alpha \;{\kappa_{1}}^{4}\eta_{3}\zeta_{3}{\mu_{3}}^{3}=0 \end{array}$$
$$\begin{array}{c} 14\;\gamma \;\kappa_{2}\zeta_{3}\eta_{2}\zeta_{4}\mu_{3}+8\;\gamma \;\kappa_{2}{\zeta_{4}}^{2}\eta_{2}\mu_{2}-105\;\alpha \;{\kappa_{2}}^{4}\eta_{2}\zeta_{3}{\mu_{3}}^{3}-500\;\alpha \;{\kappa_{1}}^{4}\eta_{2}\zeta_{4}{\mu_{3}}^{2}\mu_{2}\\ -105\;\alpha \;{\kappa_{1}}^{4}\eta_{2}\zeta_{3}{\mu_{3}}^{3}+8\;\gamma \;\kappa_{1}{\zeta_{4}}^{2}\eta_{2}\mu_{2}-500\;\alpha \;{\kappa_{2}}^{4}\eta_{2}\zeta_{4}{\mu_{3}}^{2}\mu_{2}+14\;\gamma \;\kappa_{1}\zeta_{3}\eta_{2}\zeta_{4}\mu_{3}=0 \end{array}$$
$$\begin{array}{c} 14\;\gamma \;\kappa_{1}\zeta_{3}\eta_{1}\zeta_{4}\mu_{3}-500\;\alpha \;{\kappa_{1}}^{4}\eta_{1}\zeta_{4}{\mu_{3}}^{2}\mu_{2}+14\;\gamma \;\kappa_{2}\zeta_{3}\eta_{1}\zeta_{4}\mu_{3}+8\;\gamma \;\kappa_{1}{\zeta_{4}}^{2}\eta_{1}\mu_{2}\\ +8\;\gamma \;\kappa_{2}{\zeta_{4}}^{2}\eta_{1}\mu_{2}-105\;\alpha \;{\kappa_{1}}^{4}\eta_{1}\zeta_{3}{\mu_{3}}^{3}-105\;\alpha \;{\kappa_{2}}^{4}\eta_{1}\zeta_{3}{\mu_{3}}^{3}-500\;\alpha \;{\kappa_{2}}^{4}\eta_{1}\zeta_{4}{\mu_{3}}^{2}\mu_{2}=0 \end{array}$$
$$\begin{array}{c} 4\;\gamma \;\kappa_{2}{\zeta_{4}}^{2}\eta_{4}\nu_{3}+96\;\beta \;{\kappa_{1}}^{3}\eta_{4}\nu_{3}\zeta_{4}{\mu_{3}}^{2}+4\;\gamma \;\kappa_{1}{\zeta_{4}}^{2}\eta_{4}\nu_{3}+96\;\beta \;{\kappa_{2}}^{3}\eta_{4}\nu_{3}\zeta_{4}{\mu_{3}}^{2}=0 \end{array}$$
$$\begin{array}{c} 72\;\beta \;{\kappa_{2}}^{3}\eta_{3}\nu_{3}\zeta_{4}{\mu_{3}}^{2}+3\;\gamma \;\kappa_{2}{\zeta_{4}}^{2}\eta_{3}\nu_{3}+96\;\beta \;{\kappa_{2}}^{3}\eta_{4}\nu_{2}\zeta_{4}{\mu_{3}}^{2}+96\;\beta \;{\kappa_{1}}^{3}\eta_{4}\nu_{2}\zeta_{4}{\mu_{3}}^{2}\\ +4\;\gamma \;\kappa_{1}{\zeta_{4}}^{2}\eta_{4}\nu_{2}+4\;\gamma \;\kappa_{2}{\zeta_{4}}^{2}\eta_{4}\nu_{2}+72\;\beta \;{\kappa_{1}}^{3}\eta_{3}\nu_{3}\zeta_{4}{\mu_{3}}^{2}+3\;\gamma \;\kappa_{1}{\zeta_{4}}^{2}\eta_{3}\nu_{3}=0 \end{array}$$
$$\begin{array}{c} 48\;\beta \;{\kappa_{2}}^{3}\eta_{2}\nu_{3}\zeta_{4}{\mu_{3}}^{2}+72\;\beta \;{\kappa_{2}}^{3}\eta_{3}\nu_{2}\zeta_{4}{\mu_{3}}^{2}+96\;\beta \;{\kappa_{2}}^{3}\eta_{4}\nu_{1}\zeta_{4}{\mu_{3}}^{2}+48\;\beta \;{\kappa_{1}}^{3}\eta_{2}\nu_{3}\zeta_{4}{\mu_{3}}^{2}\\ +72\;\beta \;{\kappa_{1}}^{3}\eta_{3}\nu_{2}\zeta_{4}{\mu_{3}}^{2}+96\;\beta \;{\kappa_{1}}^{3}\eta_{4}\nu_{1}\zeta_{4}{\mu_{3}}^{2}+4\;\gamma \;\kappa_{2}{\zeta_{4}}^{2}\eta_{4}\nu_{1}+3\;\gamma \;\kappa_{2}{\zeta_{4}}^{2}\eta_{3}\nu_{2}\\ -48\;\alpha \;{\kappa_{2}}^{4}\eta_{4}\zeta_{2}{\mu_{3}}^{3}-48\;\alpha \;{\kappa_{1}}^{4}\eta_{4}\zeta_{2}{\mu_{3}}^{3}+4\;\gamma \;\kappa_{1}{\zeta_{4}}^{2}\eta_{4}\nu_{1}+3\;\gamma \;\kappa_{1}{\zeta_{4}}^{2}\eta_{3}\nu_{2}\\ +2\;\gamma \;\kappa_{1}{\zeta_{4}}^{2}\eta_{2}\nu_{3}+6\;\gamma \;\kappa_{1}{\zeta_{3}}^{2}\eta_{4}\mu_{3}+8\;\gamma \;\kappa_{2}{\zeta_{4}}^{2}\eta_{4}\mu_{1}+8\;\gamma \;\kappa_{1}{\zeta_{4}}^{2}\eta_{4}\mu_{1}=0 \end{array}$$
$$\begin{array}{c} 14\;\gamma \;\kappa_{2}\zeta_{3}\eta_{4}\zeta_{4}\mu_{2}+14\;\gamma \;\kappa_{1}\zeta_{3}\eta_{4}\zeta_{4}\mu_{2}+2\;\gamma \;\kappa_{2}{\zeta_{4}}^{2}\eta_{2}\nu_{3}+6\;\gamma \;\kappa_{2}{\zeta_{3}}^{2}\eta_{4}\mu_{3}\\ -267\;\alpha \;{\kappa_{1}}^{4}\eta_{4}{\mu_{3}}^{2}\zeta_{3}\mu_{2}-432\;\alpha \;{\kappa_{1}}^{4}\eta_{4}\zeta_{4}{\mu_{3}}^{2}\mu_{1}-428\;\alpha \;{\kappa_{1}}^{4}\eta_{4}\mu_{3}\zeta_{4}{\mu_{2}}^{2}-267\;\alpha \;{\kappa_{2}}^{4}\eta_{4}{\mu_{3}}^{2}\zeta_{3}\mu_{2}\\ -432\;\alpha \;{\kappa_{2}}^{4}\eta_{4}\zeta_{4}{\mu_{3}}^{2}\mu_{1}-428\;\alpha \;{\kappa_{2}}^{4}\eta_{4}\mu_{3}\zeta_{4}{\mu_{2}}^{2}+12\;\gamma \;\kappa_{1}\zeta_{2}\eta_{4}\zeta_{4}\mu_{3}+12\;\gamma \;\kappa_{2}\zeta_{2}\eta_{4}\zeta_{4}\mu_{3}=0 \end{array}$$
$$\begin{array}{c} -48\;\alpha \;{\kappa_{2}}^{4}\eta_{3}\zeta_{2}{\mu_{3}}^{3}-48\;\alpha \;{\kappa_{1}}^{4}\eta_{3}\zeta_{2}{\mu_{3}}^{3}+2\;\gamma \;\kappa_{1}{\zeta_{4}}^{2}\eta_{2}\nu_{2}+6\;\gamma \;\kappa_{1}{\zeta_{3}}^{2}\eta_{3}\mu_{3}\\ +8\;\gamma \;\kappa_{2}{\zeta_{4}}^{2}\eta_{3}\mu_{1}+8\;\gamma \;\kappa_{1}{\zeta_{4}}^{2}\eta_{3}\mu_{1}+\gamma \;\kappa_{2}{\zeta_{4}}^{2}\eta_{1}\nu_{3}+\gamma \;\kappa_{1}{\zeta_{4}}^{2}\eta_{1}\nu_{3}\\ +4\;\gamma \;\kappa_{2}{\zeta_{4}}^{2}\eta_{4}\nu_{0}+3\;\gamma \;\kappa_{2}{\zeta_{4}}^{2}\eta_{3}\nu_{1}+2\;\gamma \;\kappa_{2}{\zeta_{4}}^{2}\eta_{2}\nu_{2}+6\;\gamma \;\kappa_{2}{\zeta_{3}}^{2}\eta_{3}\mu_{3}+4\;\gamma \;\kappa_{1}{\zeta_{4}}^{2}\eta_{4}\nu_{0}\\ +3\;\gamma \;\kappa_{1}{\zeta_{4}}^{2}\eta_{3}\nu_{1}+72\;\beta \;{\kappa_{2}}^{3}\eta_{3}\nu_{1}\zeta_{4}{\mu_{3}}^{2}+96\;\beta \;{\kappa_{2}}^{3}\eta_{4}\nu_{0}\zeta_{4}{\mu_{3}}^{2}-267\;\alpha \;{\kappa_{2}}^{4}\eta_{3}{\mu_{3}}^{2}\zeta_{3}\mu_{2}\\ -432\;\alpha \;{\kappa_{2}}^{4}\eta_{3}\zeta_{4}{\mu_{3}}^{2}\mu_{1}-428\;\alpha \;{\kappa_{2}}^{4}\eta_{3}\mu_{3}\zeta_{4}{\mu_{2}}^{2}-267\;\alpha \;{\kappa_{1}}^{4}\eta_{3}{\mu_{3}}^{2}\zeta_{3}\mu_{2}=0 \end{array}$$
$$\begin{array}{c} -432\;\alpha \;{\kappa_{1}}^{4}\eta_{3}\zeta_{4}{\mu_{3}}^{2}\mu_{1}-428\;\alpha \;{\kappa_{1}}^{4}\eta_{3}\mu_{3}\zeta_{4}{\mu_{2}}^{2}+12\;\gamma \;\kappa_{1}\zeta_{2}\eta_{3}\zeta_{4}\mu_{3}+12\;\gamma \;\kappa_{2}\zeta_{2}\eta_{3}\zeta_{4}\mu_{3}\\ +48\;\beta \;{\kappa_{2}}^{3}\eta_{2}\nu_{2}\zeta_{4}{\mu_{3}}^{2}+14\;\gamma \;\kappa_{2}\zeta_{3}\eta_{3}\zeta_{4}\mu_{2}+14\;\gamma \;\kappa_{1}\zeta_{3}\eta_{3}\zeta_{4}\mu_{2}+24\;\beta \;{\kappa_{1}}^{3}\eta_{1}\nu_{3}\zeta_{4}{\mu_{3}}^{2}\\ +48\;\beta \;{\kappa_{1}}^{3}\eta_{2}\nu_{2}\zeta_{4}{\mu_{3}}^{2}+72\;\beta \;{\kappa_{1}}^{3}\eta_{3}\nu_{1}\zeta_{4}{\mu_{3}}^{2}+96\;\beta \;{\kappa_{1}}^{3}\eta_{4}\nu_{0}\zeta_{4}{\mu_{3}}^{2}+24\;\beta \;{\kappa_{2}}^{3}\eta_{1}\nu_{3}\zeta_{4}{\mu_{3}}^{2}=0 \end{array}$$
$$\begin{array}{c} 14\;\gamma \;\kappa_{1}\zeta_{3}\eta_{2}\zeta_{4}\mu_{2}+2\;\gamma \;\kappa_{1}{\zeta_{4}}^{2}\eta_{2}\nu_{1}-48\;\alpha \;{\kappa_{2}}^{4}\eta_{2}\zeta_{2}{\mu_{3}}^{3}-48\;\alpha \;{\kappa_{1}}^{4}\eta_{2}\zeta_{2}{\mu_{3}}^{3}+6\;\gamma \;\kappa_{1}{\zeta_{3}}^{2}\eta_{2}\mu_{3}\\ +8\;\gamma \;\kappa_{1}{\zeta_{4}}^{2}\eta_{2}\mu_{1}+\gamma \;\kappa_{2}{\zeta_{4}}^{2}\eta_{1}\nu_{2}+\gamma \;\kappa_{1}{\zeta_{4}}^{2}\eta_{1}\nu_{2}+3\;\gamma \;\kappa_{2}{\zeta_{4}}^{2}\eta_{3}\nu_{0}\\ +2\;\gamma \;\kappa_{2}{\zeta_{4}}^{2}\eta_{2}\nu_{1}+8\;\gamma \;\kappa_{2}{\zeta_{4}}^{2}\eta_{2}\mu_{1}+3\;\gamma \;\kappa_{1}{\zeta_{4}}^{2}\eta_{3}\nu_{0}+6\;\gamma \;\kappa_{2}{\zeta_{3}}^{2}\eta_{2}\mu_{3}\\ +12\;\gamma \;\kappa_{1}\zeta_{2}\eta_{2}\zeta_{4}\mu_{3}+24\;\beta \;{\kappa_{1}}^{3}\eta_{1}\nu_{2}\zeta_{4}{\mu_{3}}^{2}+72\;\beta \;{\kappa_{1}}^{3}\eta_{3}\nu_{0}\zeta_{4}{\mu_{3}}^{2}+72\;\beta \;{\kappa_{2}}^{3}\eta_{3}\nu_{0}\zeta_{4}{\mu_{3}}^{2}\\ +24\;\beta \;{\kappa_{2}}^{3}\eta_{1}\nu_{2}\zeta_{4}{\mu_{3}}^{2}+48\;\beta \;{\kappa_{2}}^{3}\eta_{2}\nu_{1}\zeta_{4}{\mu_{3}}^{2}+48\;\beta \;{\kappa_{1}}^{3}\eta_{2}\nu_{1}\zeta_{4}{\mu_{3}}^{2}\\ -267\;\alpha \;{\kappa_{2}}^{4}\eta_{2}{\mu_{3}}^{2}\zeta_{3}\mu_{2}-432\;\alpha \;{\kappa_{2}}^{4}\eta_{2}\zeta_{4}{\mu_{3}}^{2}\mu_{1}-428\;\alpha \;{\kappa_{2}}^{4}\eta_{2}\mu_{3}\zeta_{4}{\mu_{2}}^{2}-267\;\alpha \;{\kappa_{1}}^{4}\eta_{2}{\mu_{3}}^{2}\zeta_{3}\mu_{2}\\ -432\;\alpha \;{\kappa_{1}}^{4}\eta_{2}\zeta_{4}{\mu_{3}}^{2}\mu_{1}-428\;\alpha \;{\kappa_{1}}^{4}\eta_{2}\mu_{3}\zeta_{4}{\mu_{2}}^{2}+12\;\gamma \;\kappa_{2}\zeta_{2}\eta_{2}\zeta_{4}\mu_{3}+14\;\gamma \;\kappa_{2}\zeta_{3}\eta_{2}\zeta_{4}\mu_{2}=0 \end{array}$$
$$\begin{array}{c} -48\;\alpha \;{\kappa_{2}}^{4}\eta_{1}\zeta_{2}{\mu_{3}}^{3}+2\;\gamma \;\kappa_{2}{\zeta_{4}}^{2}\eta_{2}\nu_{0}+24\;\beta \;{\kappa_{1}}^{3}\eta_{1}\nu_{1}\zeta_{4}{\mu_{3}}^{2}-432\;\alpha \;{\kappa_{2}}^{4}\eta_{1}\zeta_{4}{\mu_{3}}^{2}\mu_{1}\\ -428\;\alpha \;{\kappa_{2}}^{4}\eta_{1}\mu_{3}\zeta_{4}{\mu_{2}}^{2}+12\;\gamma \;\kappa_{1}\zeta_{2}\eta_{1}\zeta_{4}\mu_{3}-428\;\alpha \;{\kappa_{1}}^{4}\eta_{1}\mu_{3}\zeta_{4}{\mu_{2}}^{2}+24\;\beta \;{\kappa_{2}}^{3}\eta_{1}\nu_{1}\zeta_{4}{\mu_{3}}^{2}\\ +48\;\beta \;{\kappa_{1}}^{3}\eta_{2}\nu_{0}\zeta_{4}{\mu_{3}}^{2}-267\;\alpha \;{\kappa_{1}}^{4}\eta_{1}{\mu_{3}}^{2}\zeta_{3}\mu_{2}-48\;\alpha \;{\kappa_{1}}^{4}\eta_{1}\zeta_{2}{\mu_{3}}^{3}+\gamma \;\kappa_{1}{\zeta_{4}}^{2}\eta_{1}\nu_{1}\\ -432\;\alpha \;{\kappa_{1}}^{4}\eta_{1}\zeta_{4}{\mu_{3}}^{2}\mu_{1}-267\;\alpha \;{\kappa_{2}}^{4}\eta_{1}{\mu_{3}}^{2}\zeta_{3}\mu_{2}+48\;\beta \;{\kappa_{2}}^{3}\eta_{2}\nu_{0}\zeta_{4}{\mu_{3}}^{2}+14\;\gamma \;\kappa_{1}\zeta_{3}\eta_{1}\zeta_{4}\mu_{2}\\ +8\;\gamma \;\kappa_{1}{\zeta_{4}}^{2}\eta_{1}\mu_{1}+6\;\gamma \;\kappa_{1}{\zeta_{3}}^{2}\eta_{1}\mu_{3}+12\;\gamma \;\kappa_{2}\zeta_{2}\eta_{1}\zeta_{4}\mu_{3}+6\;\gamma \;\kappa_{2}{\zeta_{3}}^{2}\eta_{1}\mu_{3}\\ +14\;\gamma \;\kappa_{2}\zeta_{3}\eta_{1}\zeta_{4}\mu_{2}+2\;\gamma \;\kappa_{1}{\zeta_{4}}^{2}\eta_{2}\nu_{0}+8\;\gamma \;\kappa_{2}{\zeta_{4}}^{2}\eta_{1}\mu_{1}+\gamma \;\kappa_{2}{\zeta_{4}}^{2}\eta_{1}\nu_{1}=0 \end{array}$$
$$\begin{array}{c} 60\;\beta \;{\kappa_{1}}^{3}\eta_{4}\nu_{3}\zeta_{3}{\mu_{3}}^{2}+176\;\beta \;{\kappa_{2}}^{3}\eta_{4}\nu_{3}\zeta_{4}\mu_{3}\mu_{2}+8\;\gamma \;\kappa_{1}\zeta_{3}\zeta_{4}\eta_{4}\nu_{3}\\ +176\;\beta \;{\kappa_{1}}^{3}\eta_{4}\nu_{3}\zeta_{4}\mu_{3}\mu_{2}+60\;\beta \;{\kappa_{2}}^{3}\eta_{4}\nu_{3}\zeta_{3}{\mu_{3}}^{2}+8\;\gamma \;\kappa_{2}\zeta_{3}\zeta_{4}\eta_{4}\nu_{3}=0 \end{array}$$
$$\begin{array}{c} 6\;\gamma \;\kappa_{1}\zeta_{3}\zeta_{4}\eta_{3}\nu_{3}+176\;\beta \;{\kappa_{1}}^{3}\eta_{4}\nu_{2}\zeta_{4}\mu_{3}\mu_{2}+45\;\beta \;{\kappa_{1}}^{3}\eta_{3}\nu_{3}\zeta_{3}{\mu_{3}}^{2}+6\;\gamma \;\kappa_{2}\zeta_{3}\zeta_{4}\eta_{3}\nu_{3}\\ +176\;\beta \;{\kappa_{2}}^{3}\eta_{4}\nu_{2}\zeta_{4}\mu_{3}\mu_{2}+45\;\beta \;{\kappa_{2}}^{3}\eta_{3}\nu_{3}\zeta_{3}{\mu_{3}}^{2}+8\;\gamma \;\kappa_{2}\zeta_{3}\zeta_{4}\eta_{4}\nu_{2}+60\;\beta \;{\kappa_{1}}^{3}\eta_{4}\nu_{2}\zeta_{3}{\mu_{3}}^{2}\\ +8\;\gamma \;\kappa_{1}\zeta_{3}\zeta_{4}\eta_{4}\nu_{2}+60\;\beta \;{\kappa_{2}}^{3}\eta_{4}\nu_{2}\zeta_{3}{\mu_{3}}^{2}+132\;\beta \;{\kappa_{2}}^{3}\eta_{3}\nu_{3}\zeta_{4}\mu_{3}\mu_{2}+132\;\beta \;{\kappa_{1}}^{3}\eta_{3}\nu_{3}\zeta_{4}\mu_{3}\mu_{2}=0 \end{array}$$
$$\begin{array}{c} -728\;\alpha \;{\kappa_{2}}^{4}\eta_{4}\mu_{3}\mu_{2}\zeta_{4}\mu_{1}-728\;\alpha \;{\kappa_{1}}^{4}\eta_{4}\mu_{3}\mu_{2}\zeta_{4}\mu_{1}+10\;\gamma \;\kappa_{2}\zeta_{1}\eta_{4}\zeta_{4}\mu_{3}+132\;\beta \;{\kappa_{2}}^{3}\eta_{3}\nu_{2}\zeta_{4}\mu_{3}\mu_{2}\\ -15\;\alpha \;{\kappa_{1}}^{4}\eta_{4}\zeta_{1}{\mu_{3}}^{3}-120\;\alpha \;{\kappa_{1}}^{4}\eta_{4}\zeta_{4}{\mu_{2}}^{3}+6\;\gamma \;\kappa_{2}{\zeta_{3}}^{2}\eta_{4}\mu_{2}+6\;\gamma \;\kappa_{1}{\zeta_{3}}^{2}\eta_{4}\mu_{2}\\ +8\;\gamma \;\kappa_{2}{\zeta_{4}}^{2}\eta_{4}\mu_{0}+8\;\gamma \;\kappa_{1}{\zeta_{4}}^{2}\eta_{4}\mu_{0}+30\;\beta \;{\kappa_{1}}^{3}\eta_{2}\nu_{3}\zeta_{3}{\mu_{3}}^{2}+45\;\beta \;{\kappa_{1}}^{3}\eta_{3}\nu_{2}\zeta_{3}{\mu_{3}}^{2}=0 \end{array}$$
$$\begin{array}{c} 60\;\beta \;{\kappa_{1}}^{3}\eta_{4}\nu_{1}\zeta_{3}{\mu_{3}}^{2}-118\;\alpha \;{\kappa_{1}}^{4}\eta_{4}{\mu_{3}}^{2}\zeta_{2}\mu_{2}-225\;\alpha \;{\kappa_{1}}^{4}\eta_{4}{\mu_{3}}^{2}\zeta_{3}\mu_{1}-372\;\alpha \;{\kappa_{1}}^{4}\eta_{4}\zeta_{4}{\mu_{3}}^{2}\mu_{0}\\ -222\;\alpha \;{\kappa_{1}}^{4}\eta_{4}\mu_{3}\zeta_{3}{\mu_{2}}^{2}-118\;\alpha \;{\kappa_{2}}^{4}\eta_{4}{\mu_{3}}^{2}\zeta_{2}\mu_{2}-225\;\alpha \;{\kappa_{2}}^{4}\eta_{4}{\mu_{3}}^{2}\zeta_{3}\mu_{1}-372\;\alpha \;{\kappa_{2}}^{4}\eta_{4}\zeta_{4}{\mu_{3}}^{2}\mu_{0}\\ -222\;\alpha \;{\kappa_{2}}^{4}\eta_{4}\mu_{3}\zeta_{3}{\mu_{2}}^{2}+10\;\gamma \;\kappa_{1}\zeta_{2}\eta_{4}\zeta_{3}\mu_{3}+12\;\gamma \;\kappa_{1}\zeta_{2}\eta_{4}\zeta_{4}\mu_{2}+4\;\gamma \;\kappa_{1}\zeta_{3}\zeta_{4}\eta_{2}\nu_{3}\\ +6\;\gamma \;\kappa_{1}\zeta_{3}\zeta_{4}\eta_{3}\nu_{2}+8\;\gamma \;\kappa_{1}\zeta_{3}\zeta_{4}\eta_{4}\nu_{1}+4\;\gamma \;\kappa_{2}\zeta_{3}\zeta_{4}\eta_{2}\nu_{3}+176\;\beta \;{\kappa_{2}}^{3}\eta_{4}\nu_{1}\zeta_{4}\mu_{3}\mu_{2}\\ +10\;\gamma \;\kappa_{2}\zeta_{2}\eta_{4}\zeta_{3}\mu_{3}+12\;\gamma \;\kappa_{2}\zeta_{2}\eta_{4}\zeta_{4}\mu_{2}+14\;\gamma \;\kappa_{2}\zeta_{3}\eta_{4}\zeta_{4}\mu_{1}+14\;\gamma \;\kappa_{1}\zeta_{3}\eta_{4}\zeta_{4}\mu_{1}\\ +88\;\beta \;{\kappa_{1}}^{3}\eta_{2}\nu_{3}\zeta_{4}\mu_{3}\mu_{2}+10\;\gamma \;\kappa_{1}\zeta_{1}\eta_{4}\zeta_{4}\mu_{3}+132\;\beta \;{\kappa_{1}}^{3}\eta_{3}\nu_{2}\zeta_{4}\mu_{3}\mu_{2}+8\;\gamma \;\kappa_{2}\zeta_{3}\zeta_{4}\eta_{4}\nu_{1}\\ +176\;\beta \;{\kappa_{1}}^{3}\eta_{4}\nu_{1}\zeta_{4}\mu_{3}\mu_{2}+6\;\gamma \;\kappa_{2}\zeta_{3}\zeta_{4}\eta_{3}\nu_{2}-120\;\alpha \;{\kappa_{2}}^{4}\eta_{4}\zeta_{4}{\mu_{2}}^{3}-15\;\alpha \;{\kappa_{2}}^{4}\eta_{4}\zeta_{1}{\mu_{3}}^{3}\\ +88\;\beta \;{\kappa_{2}}^{3}\eta_{2}\nu_{3}\zeta_{4}\mu_{3}\mu_{2}+60\;\beta \;{\kappa_{2}}^{3}\eta_{4}\nu_{1}\zeta_{3}{\mu_{3}}^{2}+30\;\beta \;{\kappa_{2}}^{3}\eta_{2}\nu_{3}\zeta_{3}{\mu_{3}}^{2}+45\;\beta \;{\kappa_{2}}^{3}\eta_{3}\nu_{2}\zeta_{3}{\mu_{3}}^{2}=0 \end{array}$$
$$\begin{array}{c} 12\;\gamma \;\kappa_{2}\zeta_{2}\eta_{3}\zeta_{4}\mu_{2}+60\;\beta \;{\kappa_{1}}^{3}\eta_{4}\nu_{0}\zeta_{3}{\mu_{3}}^{2}+15\;\beta \;{\kappa_{2}}^{3}\eta_{1}\nu_{3}\zeta_{3}{\mu_{3}}^{2}+30\;\beta \;{\kappa_{2}}^{3}\eta_{2}\nu_{2}\zeta_{3}{\mu_{3}}^{2}\\ +45\;\beta \;{\kappa_{2}}^{3}\eta_{3}\nu_{1}\zeta_{3}{\mu_{3}}^{2}+60\;\beta \;{\kappa_{2}}^{3}\eta_{4}\nu_{0}\zeta_{3}{\mu_{3}}^{2}-118\;\alpha \;{\kappa_{2}}^{4}\eta_{3}{\mu_{3}}^{2}\zeta_{2}\mu_{2}-118\;\alpha \;{\kappa_{1}}^{4}\eta_{3}{\mu_{3}}^{2}\zeta_{2}\mu_{2}\\ -225\;\alpha \;{\kappa_{1}}^{4}\eta_{3}{\mu_{3}}^{2}\zeta_{3}\mu_{1}+88\;\beta \;{\kappa_{2}}^{3}\eta_{2}\nu_{2}\zeta_{4}\mu_{3}\mu_{2}+132\;\beta \;{\kappa_{2}}^{3}\eta_{3}\nu_{1}\zeta_{4}\mu_{3}\mu_{2}+176\;\beta \;{\kappa_{2}}^{3}\eta_{4}\nu_{0}\zeta_{4}\mu_{3}\mu_{2}\\ +44\;\beta \;{\kappa_{1}}^{3}\eta_{1}\nu_{3}\zeta_{4}\mu_{3}\mu_{2}+88\;\beta \;{\kappa_{1}}^{3}\eta_{2}\nu_{2}\zeta_{4}\mu_{3}\mu_{2}+132\;\beta \;{\kappa_{1}}^{3}\eta_{3}\nu_{1}\zeta_{4}\mu_{3}\mu_{2}+176\;\beta \;{\kappa_{1}}^{3}\eta_{4}\nu_{0}\zeta_{4}\mu_{3}\mu_{2}=0 \end{array}$$
$$\begin{array}{c} 44\;\beta \;{\kappa_{2}}^{3}\eta_{1}\nu_{3}\zeta_{4}\mu_{3}\mu_{2}-728\;\alpha \;{\kappa_{2}}^{4}\eta_{3}\mu_{3}\mu_{2}\zeta_{4}\mu_{1}-728\;\alpha \;{\kappa_{1}}^{4}\eta_{3}\mu_{3}\mu_{2}\zeta_{4}\mu_{1}-15\;\alpha \;{\kappa_{2}}^{4}\eta_{3}\zeta_{1}{\mu_{3}}^{3}\\ -120\;\alpha \;{\kappa_{2}}^{4}\eta_{3}\zeta_{4}{\mu_{2}}^{3}-15\;\alpha \;{\kappa_{1}}^{4}\eta_{3}\zeta_{1}{\mu_{3}}^{3}-120\;\alpha \;{\kappa_{1}}^{4}\eta_{3}\zeta_{4}{\mu_{2}}^{3}+6\;\gamma \;\kappa_{2}{\zeta_{3}}^{2}\eta_{3}\mu_{2}\\ +6\;\gamma \;\kappa_{1}{\zeta_{3}}^{2}\eta_{3}\mu_{2}+8\;\gamma \;\kappa_{2}{\zeta_{4}}^{2}\eta_{3}\mu_{0}+8\;\gamma \;\kappa_{1}{\zeta_{4}}^{2}\eta_{3}\mu_{0}-372\;\alpha \;{\kappa_{1}}^{4}\eta_{3}\zeta_{4}{\mu_{3}}^{2}\mu_{0}\\ -222\;\alpha \;{\kappa_{1}}^{4}\eta_{3}\mu_{3}\zeta_{3}{\mu_{2}}^{2}-225\;\alpha \;{\kappa_{2}}^{4}\eta_{3}{\mu_{3}}^{2}\zeta_{3}\mu_{1}-372\;\alpha \;{\kappa_{2}}^{4}\eta_{3}\zeta_{4}{\mu_{3}}^{2}\mu_{0}=0 \end{array}$$
$$\begin{array}{c} -222\;\alpha \;{\kappa_{2}}^{4}\eta_{3}\mu_{3}\zeta_{3}{\mu_{2}}^{2}+12\;\gamma \;\kappa_{1}\zeta_{2}\eta_{3}\zeta_{4}\mu_{2}+2\;\gamma \;\kappa_{1}\zeta_{3}\zeta_{4}\eta_{1}\nu_{3}+4\;\gamma \;\kappa_{1}\zeta_{3}\zeta_{4}\eta_{2}\nu_{2}\\ +6\;\gamma \;\kappa_{1}\zeta_{3}\zeta_{4}\eta_{3}\nu_{1}+8\;\gamma \;\kappa_{1}\zeta_{3}\zeta_{4}\eta_{4}\nu_{0}+10\;\gamma \;\kappa_{2}\zeta_{1}\eta_{3}\zeta_{4}\mu_{3}+\gamma \;\kappa_{1}\zeta_{2}\eta_{3}\zeta_{3}\mu_{3}\\ +10\;\gamma \;\kappa_{2}\zeta_{2}\eta_{3}\zeta_{3}\mu_{3}+15\;\beta \;{\kappa_{1}}^{3}\eta_{1}\nu_{3}\zeta_{3}{\mu_{3}}^{2}+30\;\beta \;{\kappa_{1}}^{3}\eta_{2}\nu_{2}\zeta_{3}{\mu_{3}}^{2}+45\;\beta \;{\kappa_{1}}^{3}\eta_{3}\nu_{1}\zeta_{3}{\mu_{3}}^{2}\\ +14\;\gamma \;\kappa_{2}\zeta_{3}\eta_{3}\zeta_{4}\mu_{1}+14\;\gamma \;\kappa_{1}\zeta_{3}\eta_{3}\zeta_{4}\mu_{1}+10\;\gamma \;\kappa_{1}\zeta_{1}\eta_{3}\zeta_{4}\mu_{3}+8\;\gamma \;\kappa_{2}\zeta_{3}\zeta_{4}\eta_{4}\nu_{0}=0 \end{array}$$
$$\begin{array}{c} 6\;\gamma \;\kappa_{2}\zeta_{3}\zeta_{4}\eta_{3}\nu_{1}+4\;\gamma \;\kappa_{2}\zeta_{3}\zeta_{4}\eta_{2}\nu_{2}+2\;\gamma \;\kappa_{2}\zeta_{3}\zeta_{4}\eta_{1}\nu_{3}=0 \end{array}$$
$$\begin{array}{c} 132\;\beta \;{\kappa_{1}}^{3}\eta_{3}\nu_{0}\zeta_{4}\mu_{3}\mu_{2}-120\;\alpha \;{\kappa_{2}}^{4}\eta_{2}\zeta_{4}{\mu_{2}}^{3}-15\;\alpha \;{\kappa_{2}}^{4}\eta_{2}\zeta_{1}{\mu_{3}}^{3}+88\;\beta \;{\kappa_{1}}^{3}\eta_{2}\nu_{1}\zeta_{4}\mu_{3}\mu_{2}\\ -372\;\alpha \;{\kappa_{1}}^{4}\eta_{2}\zeta_{4}{\mu_{3}}^{2}\mu_{0}-222\;\alpha \;{\kappa_{1}}^{4}\eta_{2}\mu_{3}\zeta_{3}{\mu_{2}}^{2}+132\;\beta \;{\kappa_{2}}^{3}\eta_{3}\nu_{0}\zeta_{4}\mu_{3}\mu_{2}+30\;\beta \;{\kappa_{2}}^{3}\eta_{2}\nu_{1}\zeta_{3}{\mu_{3}}^{2}\\ +88\;\beta \;{\kappa_{2}}^{3}\eta_{2}\nu_{1}\zeta_{4}\mu_{3}\mu_{2}+8\;\gamma \;\kappa_{2}{\zeta_{4}}^{2}\eta_{2}\mu_{0}-15\;\alpha \;{\kappa_{1}}^{4}\eta_{2}\zeta_{1}{\mu_{3}}^{3}-120\;\alpha \;{\kappa_{1}}^{4}\eta_{2}\zeta_{4}{\mu_{2}}^{3}\\ -118\;\alpha \;{\kappa_{1}}^{4}\eta_{2}{\mu_{3}}^{2}\zeta_{2}\mu_{2}-225\;\alpha \;{\kappa_{1}}^{4}\eta_{2}{\mu_{3}}^{2}\zeta_{3}\mu_{1}-728\;\alpha \;{\kappa_{1}}^{4}\eta_{2}\mu_{3}\mu_{2}\zeta_{4}\mu_{1}-728\;\alpha \;{\kappa_{2}}^{4}\eta_{2}\mu_{3}\mu_{2}\zeta_{4}\mu_{1}=0 \end{array}$$
$$\begin{array}{c} 2\;\gamma \;\kappa_{2}\zeta_{3}\zeta_{4}\eta_{1}\nu_{2}+15\;\beta \;{\kappa_{1}}^{3}\eta_{1}\nu_{2}\zeta_{3}{\mu_{3}}^{2}+6\;\gamma \;\kappa_{1}\zeta_{3}\zeta_{4}\eta_{3}\nu_{0}+10\;\gamma \;\kappa_{1}\zeta_{2}\eta_{2}\zeta_{3}\mu_{3}\\ +6\;\gamma \;\kappa_{2}\zeta_{3}\zeta_{4}\eta_{3}\nu_{0}+12\;\gamma \;\kappa_{1}\zeta_{2}\eta_{2}\zeta_{4}\mu_{2}+2\;\gamma \;\kappa_{1}\zeta_{3}\zeta_{4}\eta_{1}\nu_{2}+4\;\gamma \;\kappa_{1}\zeta_{3}\zeta_{4}\eta_{2}\nu_{1}\\ +10\;\gamma \;\kappa_{1}\zeta_{1}\eta_{2}\zeta_{4}\mu_{3}+6\;\gamma \;\kappa_{1}{\zeta_{3}}^{2}\eta_{2}\mu_{2}+10\;\gamma \;\kappa_{2}\zeta_{1}\eta_{2}\zeta_{4}\mu_{3}+10\;\gamma \;\kappa_{2}\zeta_{2}\eta_{2}\zeta_{3}\mu_{3}\\ +4\;\gamma \;\kappa_{2}\zeta_{3}\zeta_{4}\eta_{2}\nu_{1}+6\;\gamma \;\kappa_{2}{\zeta_{3}}^{2}\eta_{2}\mu_{2}+12\;\gamma \;\kappa_{2}\zeta_{2}\eta_{2}\zeta_{4}\mu_{2}+14\;\gamma \;\kappa_{2}\zeta_{3}\eta_{2}\zeta_{4}\mu_{1}\\ -225\;\alpha \;{\kappa_{2}}^{4}\eta_{2}{\mu_{3}}^{2}\zeta_{3}\mu_{1}+30\;\beta \;{\kappa_{1}}^{3}\eta_{2}\nu_{1}\zeta_{3}{\mu_{3}}^{2}-118\;\alpha \;{\kappa_{2}}^{4}\eta_{2}{\mu_{3}}^{2}\zeta_{2}\mu_{2}+15\;\beta \;{\kappa_{2}}^{3}\eta_{1}\nu_{2}\zeta_{3}{\mu_{3}}^{2}\\ +45\;\beta \;{\kappa_{1}}^{3}\eta_{3}\nu_{0}\zeta_{3}{\mu_{3}}^{2}+45\;\beta \;{\kappa_{2}}^{3}\eta_{3}\nu_{0}\zeta_{3}{\mu_{3}}^{2}+44\;\beta \;{\kappa_{2}}^{3}\eta_{1}\nu_{2}\zeta_{4}\mu_{3}\mu_{2}+44\;\beta \;{\kappa_{1}}^{3}\eta_{1}\nu_{2}\zeta_{4}\mu_{3}\mu_{2}\\ +8\;\gamma \;\kappa_{1}{\zeta_{4}}^{2}\eta_{2}\mu_{0}+14\;\gamma \;\kappa_{1}\zeta_{3}\eta_{2}\zeta_{4}\mu_{1}-222\;\alpha \;{\kappa_{2}}^{4}\eta_{2}\mu_{3}\zeta_{3}{\mu_{2}}^{2}=0 \end{array}$$
$$\begin{array}{c} -728\;\alpha \;{\kappa_{1}}^{4}\eta_{1}\mu_{3}\mu_{2}\zeta_{4}\mu_{1}+44\;\beta \;{\kappa_{2}}^{3}\eta_{1}\nu_{1}\zeta_{4}\mu_{3}\mu_{2}+88\;\beta \;{\kappa_{2}}^{3}\eta_{2}\nu_{0}\zeta_{4}\mu_{3}\mu_{2}+44\;\beta \;{\kappa_{1}}^{3}\eta_{1}\nu_{1}\zeta_{4}\mu_{3}\mu_{2}\\ +88\;\beta \;{\kappa_{1}}^{3}\eta_{2}\nu_{0}\zeta_{4}\mu_{3}\mu_{2}-728\;\alpha \;{\kappa_{2}}^{4}\eta_{1}\mu_{3}\mu_{2}\zeta_{4}\mu_{1}-15\;\alpha \;{\kappa_{2}}^{4}\eta_{1}\zeta_{1}{\mu_{3}}^{3}-120\;\alpha \;{\kappa_{2}}^{4}\eta_{1}\zeta_{4}{\mu_{2}}^{3}\\ -15\;\alpha \;{\kappa_{1}}^{4}\eta_{1}\zeta_{1}{\mu_{3}}^{3}-120\;\alpha \;{\kappa_{1}}^{4}\eta_{1}\zeta_{4}{\mu_{2}}^{3}+8\;\gamma \;\kappa_{2}{\zeta_{4}}^{2}\eta_{1}\mu_{0}+6\;\gamma \;\kappa_{2}{\zeta_{3}}^{2}\eta_{1}\mu_{2}\\ +6\;\gamma \;\kappa_{1}{\zeta_{3}}^{2}\eta_{1}\mu_{2}+8\;\gamma \;\kappa_{1}{\zeta_{4}}^{2}\eta_{1}\mu_{0}-118\;\alpha \;{\kappa_{1}}^{4}\eta_{1}{\mu_{3}}^{2}\zeta_{2}\mu_{2}+15\;\beta \;{\kappa_{1}}^{3}\eta_{1}\nu_{1}\zeta_{3}{\mu_{3}}^{2}=0 \end{array}$$
$$\begin{array}{c} 30\;\beta \;{\kappa_{1}}^{3}\eta_{2}\nu_{0}\zeta_{3}{\mu_{3}}^{2}+15\;\beta \;{\kappa_{2}}^{3}\eta_{1}\nu_{1}\zeta_{3}{\mu_{3}}^{2}+30\;\beta \;{\kappa_{2}}^{3}\eta_{2}\nu_{0}\zeta_{3}{\mu_{3}}^{2}-225\;\alpha \;{\kappa_{1}}^{4}\eta_{1}{\mu_{3}}^{2}\zeta_{3}\mu_{1}\\ -372\;\alpha \;{\kappa_{1}}^{4}\eta_{1}\zeta_{4}{\mu_{3}}^{2}\mu_{0}-222\;\alpha \;{\kappa_{1}}^{4}\eta_{1}\mu_{3}\zeta_{3}{\mu_{2}}^{2}-118\;\alpha \;{\kappa_{2}}^{4}\eta_{1}{\mu_{3}}^{2}\zeta_{2}\mu_{2}\\ -225\;\alpha \;{\kappa_{2}}^{4}\eta_{1}{\mu_{3}}^{2}\zeta_{3}\mu_{1}-372\;\alpha \;{\kappa_{2}}^{4}\eta_{1}\zeta_{4}{\mu_{3}}^{2}\mu_{0}-222\;\alpha \;{\kappa_{2}}^{4}\eta_{1}\mu_{3}\zeta_{3}{\mu_{2}}^{2}+2\;\gamma \;\kappa_{1}\zeta_{3}\zeta_{4}\eta_{1}\nu_{1}\\ +4\;\gamma \;\kappa_{1}\zeta_{3}\zeta_{4}\eta_{2}\nu_{0}+10\;\gamma \;\kappa_{2}\zeta_{2}\eta_{1}\zeta_{3}\mu_{3}+12\;\gamma \;\kappa_{2}\zeta_{2}\eta_{1}\zeta_{4}\mu_{2}+14\;\gamma \;\kappa_{2}\zeta_{3}\eta_{1}\zeta_{4}\mu_{1}\\ +14\;\gamma \;\kappa_{1}\zeta_{3}\eta_{1}\zeta_{4}\mu_{1}+10\;\gamma \;\kappa_{2}\zeta_{1}\eta_{1}\zeta_{4}\mu_{3}+12\;\gamma \;\kappa_{1}\zeta_{2}\eta_{1}\zeta_{4}\mu_{2}+10\;\gamma \;\kappa_{1}\zeta_{2}\eta_{1}\zeta_{3}\mu_{3}\\ +10\;\gamma \;\kappa_{1}\zeta_{1}\eta_{1}\zeta_{4}\mu_{3}+4\;\gamma \;\kappa_{2}\zeta_{3}\zeta_{4}\eta_{2}\nu_{0}+2\;\gamma \;\kappa_{2}\zeta_{3}\zeta_{4}\eta_{1}\nu_{1}=0 \end{array}$$
$$\begin{array}{c} 60\;\beta \;{\kappa_{1}}^{3}\zeta_{4}\mu_{3}\eta_{3}{\nu_{3}}^{2}+60\;\beta \;{\kappa_{2}}^{3}\zeta_{4}\mu_{3}\eta_{3}{\nu_{3}}^{2}+176\;\beta \;{\kappa_{2}}^{3}\zeta_{4}\mu_{3}\eta_{4}\nu_{3}\nu_{2}\\ +8\;\gamma \;\kappa_{1}\eta_{3}\eta_{4}\zeta_{4}\mu_{3}+176\;\beta \;{\kappa_{1}}^{3}\zeta_{4}\mu_{3}\eta_{4}\nu_{3}\nu_{2}+8\;\gamma \;\kappa_{2}\eta_{3}\eta_{4}\zeta_{4}\mu_{3}=0 \end{array}$$
$$\begin{array}{c} 72\;\beta \;{\kappa_{2}}^{3}\zeta_{3}\mu_{3}\eta_{4}{\nu_{3}}^{2}+4\;\gamma \;\kappa_{1}{\eta_{4}}^{2}\zeta_{4}\mu_{2}+3\;\gamma \;\kappa_{1}{\eta_{4}}^{2}\zeta_{3}\mu_{3}+96\;\beta \;{\kappa_{1}}^{3}\zeta_{4}\mu_{2}\eta_{4}{\nu_{3}}^{2}\\ +4\;\gamma \;\kappa_{2}{\eta_{4}}^{2}\zeta_{4}\mu_{2}+3\;\gamma \;\kappa_{2}{\eta_{4}}^{2}\zeta_{3}\mu_{3}+72\;\beta \;{\kappa_{1}}^{3}\zeta_{3}\mu_{3}\eta_{4}{\nu_{3}}^{2}+96\;\beta \;{\kappa_{2}}^{3}\zeta_{4}\mu_{2}\eta_{4}{\nu_{3}}^{2}=0 \end{array}$$
$$\begin{array}{c} 2416\;{\kappa_{2}}^{5}{\mu_{1}}^{4}+2416\;{\kappa_{1}}^{5}{\mu_{1}}^{4}+\omega_{1}-44\;\delta \;{\kappa_{1}}^{3}{\mu_{1}}^{2}-44\;\delta \;{\kappa_{2}}^{3}{\mu_{1}}^{2}+\omega_{2}=0 \end{array}$$
$$\begin{array}{c} -48\;\delta \;{\kappa_{1}}^{3}{\mu_{1}}^{2}+5280\;{\kappa_{1}}^{5}{\mu_{1}}^{4}-48\;\delta \;{\kappa_{2}}^{3}{\mu_{1}}^{2}+5280\;{\kappa_{2}}^{5}{\mu_{1}}^{4}=0 \end{array}$$

Thanks to the computer algebra system MAPLE, solutions of the system (Equation 27) can be derived. We note that this is performed step by step starting from the simplest equation of the system of algebraic equations and then moving in the direction of solving of the more complicated equations. There are many variants of solutions, which can be obtaining by the computational software, but our goal is to express the coefficients of the solved equation and the coefficients of its solution by the coefficients of the simple equations and the coefficients in the solutions of the simple equations, so far as it is possible.

For the special case $\mu_{0}=\mu_{2}=0$ and $\nu_{0}=\nu_{2}=0$, one non-trivial solution of the system (Equation 27), presenting the relationships between the coefficients of the solved equation and the coefficients of the simple equations and their solutions, is: (28)$$\begin{array}{c} \zeta_{0}=\zeta_{1}=\zeta_{3}=0,\;\zeta_{2}=\zeta_{4}=1,\;\eta_{0}=\eta_{1}=\eta_{3}=0,\;\eta_{2}=\eta_{4}=1,\;\mu_{3}=-\mu_{1},\;\nu_{3}=-\nu_{1},\\ \alpha =120\;\frac{\left({\kappa_{1}}^{5}+{\kappa_{2}}^{5}\right){\nu_{1}}^{2}}{{\kappa_{2}}^{4}+{\kappa_{1}}^{4}},\;\beta =-120\;\frac{\left({\kappa_{1}}^{4}-\kappa_{2}{\kappa_{1}}^{3}+{\kappa_{2}}^{2}{\kappa_{1}}^{2}-{\kappa_{2}}^{3}\kappa_{1}+{\kappa_{2}}^{4}\right){\nu_{1}}^{2}}{{\kappa_{1}}^{2}-\kappa_{2}\kappa_{1}+{\kappa_{2}}^{2}},\\ \gamma =2880\;\left({\kappa_{1}}^{4}-\kappa_{2}{\kappa_{1}}^{3}+{\kappa_{2}}^{2}{\kappa_{1}}^{2}-{\kappa_{2}}^{3}\kappa_{1}+{\kappa_{2}}^{4}\right){\nu_{1}}^{4},\;\omega_{1}=1984\left(\kappa_{1}^{5}+\kappa_{2}^{5}\right)\nu_{1}^{4}-\omega_{2},\\ \delta =10\;\frac{10\;{\kappa_{1}}^{4}{\nu_{1}}^{2}-10\;\kappa_{2}{\kappa_{1}}^{3}{\nu_{1}}^{2}+10\;{\kappa_{2}}^{2}{\kappa_{1}}^{2}{\nu_{1}}^{2}-10\;{\kappa_{2}}^{3}\kappa_{1}{\nu_{1}}^{2}+10\;{\kappa_{2}}^{4}{\nu_{1}}^{2}}{{\kappa_{1}}^{2}-\kappa_{2}\kappa_{1}+{\kappa_{2}}^{2}},\\ \epsilon =-4800\;{\kappa_{1}}^{4}{\nu_{1}}^{2}+4800\;\kappa_{2}{\kappa_{1}}^{3}{\nu_{1}}^{2}-4800\;{\kappa_{2}}^{2}{\kappa_{1}}^{2}{\nu_{1}}^{2}+4800\;{\kappa_{2}}^{3}\kappa_{1}{\nu_{1}}^{2}-4800\;{\kappa_{2}}^{4}{\nu_{1}}^{2}, \end{array}$$
where $\kappa_{1},\kappa_{2},\mu_{1},\nu_{1}$ and $\omega_{2}$ are free parameters.

Then, the solution of Equation (Equation 1) has the following form:(29)$$u\left(\xi_{1},\xi_{2}\right)=1+f_{1}{\left(\xi_{1}\right)}^{2}+f_{1}{\left(\xi_{1}\right)}^{4}+f_{2}{\left(\xi_{2}\right)}^{2}+f_{2}{\left(\xi_{2}\right)}^{4}$$
where the simple equations are ODEs of Bernoulli kind, i.e.,
(30)$$\frac{df_{1}}{d\xi_{1}}=\mu_{1}f_{1}-\mu_{1}f_{1}^{3},\;\;\frac{df_{2}}{d\xi_{2}}=\nu_{1}f_{2}-\nu_{1}f_{2}^{3}$$

The solutions of Equation (Equation 30) can be presented by the special functions *V* in the following way:(31)$$f_{1}=V_{0,\mu_{1},0,-\mu_{1}}\left(\xi_{1};1,1,3\right),\;\;f_{2}=V_{0,\nu_{1},0,-\nu_{1}}\left(\xi_{2};1,1,3\right)$$

Then, the solution of Equation (Equation 29) can be rewritten as
(32)$$\begin{array}{c} u\left(\xi_{1},\xi_{2}\right)=1+V_{0,\mu_{1},0,\mu_{1}}^{2}\left(\xi_{1};1,1,3\right)+V_{0,\mu_{1},0,-\mu_{1}}^{4}\left(\xi_{1};1,1,3\right)\\ +V_{0,\nu_{1},0,-\nu_{1}}^{2}\left(\xi_{2};1,1,3\right)+V_{0,\nu_{1},0,-\nu_{1}}^{4}\left(\xi_{2};1,1,3\right) \end{array}$$

We note that in the context of the general solution of the Bernoulli ordinary differential equation (see Equations (Equation 11) and (Equation 12)), the special functions given above reduce to the following specific forms:(33)$$\begin{array}{c} V_{0,\mu_{1},0,-\mu_{1}}\left(\xi_{1};1,1,3\right)=\sqrt{\frac{\mu_{1}\exp\left(2\mu_{1}\xi_{1}\right)}{1-\mu_{1}\exp\left(2\mu_{1}\xi_{1}\right)}},\\ V_{0,\nu_{1},0,-\nu_{1}}\left(\xi_{1};1,1,3\right)=\sqrt{\frac{\nu_{1}\exp\left(2\nu_{1}\xi_{1}\right)}{1-\nu_{1}\exp\left(2\mu_{1}\xi_{1}\right)}} \end{array}$$
for $\mu_{1}>0$ and $\nu_{1}>0$ and
(34)$$\begin{array}{c} V_{0,\mu_{1},0,-\mu_{1}}\left(\xi_{1};1,1,3\right)=\sqrt{\frac{\mu_{1}\exp\left(2\mu_{1}\xi_{1}\right)}{1+\mu_{1}\exp\left(2\mu_{1}\xi_{1}\right)}},\\ V_{0,\nu_{1},0,-\nu_{1}}\left(\xi_{1};1,1,3\right)=\sqrt{\frac{\nu_{1}\exp\left(2\nu_{1}\xi_{1}\right)}{1+\nu_{1}\exp\left(2\mu_{1}\xi_{1}\right)}} \end{array}$$
for $\mu_{1}<0$ and $\nu_{1}<0$, where
(35)$$\xi_{1}=\kappa_{1}x+\left(1984\left(\kappa_{1}^{5}+\kappa_{2}^{5}\right)\nu_{1}^{4}-\omega_{2}\right)t,\;\;\xi_{2}=\kappa_{2}x+\omega_{2}t$$Finally, for this specific case, the travelling-wave solutions of Equation (Equation 1) can be presented as follows:(36)$$\begin{array}{c} u\left(x,t\right)=1+\frac{\mu_{1}\exp\left[2\mu_{1}\left(\kappa_{1}x-\left(\omega_{2}-1984\left(\kappa_{1}^{5}+\kappa_{2}^{5}\right)\nu_{1}^{4}\right)t\right)\right]}{1-\mu_{1}\exp\left[2\mu_{1}\left(\kappa_{1}x-\left(\omega_{2}-1984\left(\kappa_{1}^{5}+\kappa_{2}^{5}\right)\nu_{1}^{4}\right)t\right)\right]}\\ +{\left(\frac{\mu_{1}\exp\left[2\mu_{1}\left(\kappa_{1}x-\left(\omega_{2}-1984\left(\kappa_{1}^{5}+\kappa_{2}^{5}\right)\nu_{1}^{4}\right)t\right)\right]}{1-\mu_{1}\exp\left[2\mu_{1}\left(\kappa_{1}x-\left(\omega_{2}-1984\left(\kappa_{1}^{5}+\kappa_{2}^{5}\right)\nu_{1}^{4}\right)t\right)\right]}\right)}^{2}\\ +\frac{\nu_{1}\exp\left[2\nu_{1}\left(\kappa_{2}x+\omega_{2}t\right)\right]}{1-\nu_{1}\exp[2\nu_{1}\left(\kappa_{2}x+\omega_{2}t\right)]}+{\left(\frac{\nu_{1}\exp[2\nu_{1}\left(\kappa_{2}x+\omega_{2}t\right)]}{1-\nu_{1}\exp\left[2\nu_{1}\left(\kappa_{2}x+\omega_{2}t\right)\right]}\right)}^{2} \end{array}$$
for $\mu_{1}>0$ and $\nu_{1}>0$ and
(37)$$\begin{array}{c} u\left(x,t\right)=1+\frac{\mu_{1}\exp\left[2\mu_{1}\left(\kappa_{1}x-\left(\omega_{2}-1984\left(\kappa_{1}^{5}+\kappa_{2}^{5}\right)\nu_{1}^{4}\right)t\right)\right]}{1+\mu_{1}\exp\left[2\mu_{1}\left(\kappa_{1}x-\left(\omega_{2}-1984\left(\kappa_{1}^{5}+\kappa_{2}^{5}\right)\nu_{1}^{4}\right)t\right)\right]}\\ +{\left(\frac{\mu_{1}\exp\left[2\mu_{1}\left(\kappa_{1}x-\left(\omega_{2}-1984\left(\kappa_{1}^{5}+\kappa_{2}^{5}\right)\nu_{1}^{4}\right)t\right)\right]}{1+\mu_{1}\exp\left[2\mu_{1}\left(\kappa_{1}x-\left(\omega_{2}-1984\left(\kappa_{1}^{5}+\kappa_{2}^{5}\right)\nu_{1}^{4}\right)t\right)\right]}\right)}^{2}\\ +\frac{\nu_{1}\exp\left[2\nu_{1}\left(\kappa_{2}x+\omega_{2}t\right)\right]}{1+\nu_{1}\exp\left[2\nu_{1}\left(\kappa_{2}x+\omega_{2}t\right)\right]}+{\left(\frac{\nu_{1}\exp\left[2\nu_{1}\left(\kappa_{2}x+\omega_{2}t\right)\right]}{1+\nu_{1}\exp\left[2\nu_{1}\left(\kappa_{2}x+\omega_{2}t\right)\right]}\right)}^{2} \end{array}$$
for $\mu_{1}<0$ and $\nu_{1}<0$.

### 3.2. Case $m_{1}=2,\;m_{2}=3$

For this case, $\mu_{3}=0$. Then, according to balance equation for $n_{1}$ (see Equation (Equation 23)), $\zeta_{3}=0$, too, i.e., $n_{1}=2$. The function *u* is presented as follows:(38)$$u\left(\xi_{1},\xi_{2}\right)=1+\sum_{i_{1}=0}^{2}\zeta_{i_{1}}{\left[f_{1}\left(\xi_{1}\right)\right]}^{i_{1}}+\sum_{i_{2}=0}^{4}\eta_{i_{2}}{\left[f_{2}\left(\xi_{2}\right)\right]}^{i_{2}},$$
where
(39)$$\frac{df_{1}}{d\xi_{1}}=\mu_{0}+\mu_{1}f_{1}+\mu_{2}f_{1}^{2},\;\;\frac{df_{2}}{d\xi_{2}}=\nu_{0}+\nu_{1}f_{2}+\nu_{2}f_{2}^{2}+\nu_{3}f_{2}^{3}$$

For this case, one non-trivial solution of the system (Equation 27) is: (40)$$\begin{array}{c} \zeta_{0}=\zeta_{2}=1,\;\zeta_{1}=\frac{1}{8}\;\frac{\mu_{1}\Omega_{1}}{16\;\mu_{0}+\Omega_{2}},\;\eta_{0}=\eta_{3}=\eta_{4}=1,\;\eta_{1}=\frac{1}{2}\;\eta_{2}-\frac{1}{8},\;\mu_{2}=8\;\frac{16\;\mu_{0}+\Omega_{2}}{\Omega_{1}},\\ \nu_{0}=\frac{\left(\Omega_{2}+\mu_{0}\right)\left(-5+16\;\eta_{2}\right)}{16\Omega_{1}},\;\nu_{1}=\frac{1}{4}\;\frac{\left(16\;\eta_{2}-3\right)\left(16\;\mu_{0}+\Omega_{2}\right)}{\Omega_{1}},\;\nu_{2}=3\;\frac{16\;\mu_{0}+\Omega_{2}}{\Omega_{1}},\\ \nu_{3}=4\;\frac{16\;\mu_{0}+\Omega_{2}}{\Omega_{1}},\;\alpha =1920\;\frac{\left({\kappa_{1}}^{5}+{\kappa_{2}}^{5}\right){\left(16\;\mu_{0}+\Omega_{2}\right)}^{2}}{\left({\kappa_{1}}^{4}+{\kappa_{2}}^{4}\right){\Omega_{1}}^{2}},\\ \beta =-1920\;\frac{\left({\kappa_{1}}^{4}-\kappa_{2}{\kappa_{1}}^{3}+{\kappa_{2}}^{2}{\kappa_{1}}^{2}-{\kappa_{2}}^{3}\kappa_{1}+{\kappa_{2}}^{4}\right){\left(16\;\mu_{0}+\Omega_{2}\right)}^{2}}{\Omega_{1}^{2}\left({\kappa_{1}}^{2}-\kappa_{2}\kappa_{1}+{\kappa_{2}}^{2}\right)},\\ \gamma =737280\frac{\left(\kappa_{1}^{4}-\kappa_{2}\kappa_{1}^{3}+\kappa_{2}^{2}\kappa_{1}^{2}-\kappa_{2}^{3}\kappa_{1}+\kappa_{2}^{4}\right){\left(16\;\mu_{0}+\Omega_{2}\right)}^{4}}{\Omega_{1}^{4}},\\ \delta =-\frac{15}{2{\Omega_{1}}^{2}\left({\kappa_{1}}^{2}-\kappa_{2}\kappa_{1}+{\kappa_{2}}^{2}\right)}\;(-6957\;\kappa_{1}{\kappa_{2}}^{3}{\mu_{1}}^{2}-6957\;{\kappa_{1}}^{3}\kappa_{2}{\mu_{1}}^{2}+6957\;{\kappa_{1}}^{2}{\kappa_{2}}^{2}{\mu_{1}}^{2}\\ -16384\;\kappa_{1}^{4}\mu_{0}^{2}\eta_{2}+32768\;{\kappa_{1}}^{4}{\mu_{0}}^{2}{\eta_{2}}^{2}-37248\;{\kappa_{1}}^{4}{\mu_{1}}^{2}\eta_{2}+50240\;{\kappa_{1}}^{4}{\mu_{1}}^{2}{\eta_{2}}^{2}\\ -1024\;{\kappa_{1}}^{4}{\mu_{1}}^{2}{\eta_{2}}^{3}+391168\;\kappa_{2}{\kappa_{1}}^{3}{\mu_{0}}^{2}-391168\;{\kappa_{2}}^{2}{\kappa_{1}}^{2}{\mu_{0}}^{2}+391168\;\kappa_{1}{\kappa_{2}}^{3}{\mu_{0}}^{2}\\ -16384\;{\kappa_{2}}^{4}{\mu_{0}}^{2}\eta_{2}+32768\;{\kappa_{2}}^{4}{\mu_{0}}^{2}{\eta_{2}}^{2}-37248\;{\kappa_{2}}^{4}{\mu_{1}}^{2}\eta_{2}+50240\;{\kappa_{2}}^{4}{\mu_{1}}^{2}{\eta_{2}}^{2}\\ -1024\;{\kappa_{2}}^{4}{\mu_{1}}^{2}{\eta_{2}}^{3}+6957\;{\kappa_{2}}^{4}{\mu_{1}}^{2}+6957\;{\kappa_{1}}^{4}{\mu_{1}}^{2}-391168\;{\kappa_{2}}^{4}{\mu_{0}}^{2}\\ -391168\;{\kappa_{1}}^{4}{\mu_{0}}^{2}+37248\;\kappa_{1}{\kappa_{2}}^{3}{\mu_{1}}^{2}\eta_{2}-50240\;\kappa_{1}{\kappa_{2}}^{3}{\mu_{1}}^{2}{\eta_{2}}^{2}+1024\;\kappa_{1}{\kappa_{2}}^{3}{\mu_{1}}^{2}{\eta_{2}}^{3}\\ -32768\;{\kappa_{1}}^{3}\kappa_{2}{\eta_{2}}^{2}{\mu_{0}}^{2}+16384\;{\kappa_{1}}^{3}\kappa_{2}{\mu_{0}}^{2}\eta_{2}+37248\;{\kappa_{1}}^{3}\kappa_{2}{\mu_{1}}^{2}\eta_{2}-50240\;{\kappa_{1}}^{3}\kappa_{2}{\mu_{1}}^{2}{\eta_{2}}^{2}\\ +1024\;{\kappa_{1}}^{3}\kappa_{2}{\mu_{1}}^{2}{\eta_{2}}^{3}-37248\;{\kappa_{1}}^{2}{\kappa_{2}}^{2}{\mu_{1}}^{2}\eta_{2}+50240\;{\kappa_{1}}^{2}{\kappa_{2}}^{2}{\mu_{1}}^{2}{\eta_{2}}^{2}-1024\;{\kappa_{1}}^{2}{\kappa_{2}}^{2}{\mu_{1}}^{2}{\eta_{2}}^{3}\\ -16384\;{\kappa_{1}}^{2}{\kappa_{2}}^{2}{\mu_{0}}^{2}\eta_{2}+32768\;{\kappa_{1}}^{2}{\kappa_{2}}^{2}{\mu_{0}}^{2}{\eta_{2}}^{2}+16384\;\kappa_{1}{\kappa_{2}}^{3}{\mu_{0}}^{2}\eta_{2}-32768\;\kappa_{1}{\kappa_{2}}^{3}{\mu_{0}}^{2}{\eta_{2}}^{2}\\ -24448\;{\kappa_{2}}^{4}\mu_{0}\Omega_{2}-1024\;{\kappa_{1}}^{4}\Omega_{2}\eta_{2}-1024\;{\kappa_{2}}^{4}\mu_{0}\Omega_{2}\eta_{2}+2048\;{\kappa_{2}}^{4}\mu_{0}\Omega_{2}{\eta_{2}}^{2}\\ +2048\;{\kappa_{1}}^{4}\mu_{0}\Omega_{2}{\eta_{2}}^{2}+24448\;\kappa_{2}{\kappa_{1}}^{3}\mu_{0}\Omega_{2}-24448\;{\kappa_{2}}^{2}{\kappa_{1}}^{2}\mu_{0}\Omega_{2}+24448\;\kappa_{1}{\kappa_{2}}^{3}\mu_{0}\Omega_{2}\\ -2048\;{\kappa_{1}}^{3}\kappa_{2}{\eta_{2}}^{2}\mu_{0}\Omega_{2}+1024\;\kappa_{2}{\kappa_{1}}^{3}\mu_{0}\Omega_{2}\eta_{2}-1024\;{\kappa_{2}}^{2}{\kappa_{1}}^{2}\mu_{0}\Omega_{2}\eta_{2}+2048\;{\kappa_{2}}^{2}{\kappa_{1}}^{2}\mu_{0}\Omega_{2}{\eta_{2}}^{2}\\ +1024\;\kappa_{1}{\kappa_{2}}^{3}\mu_{0}\Omega_{2}\eta_{2}-2048\;\kappa_{1}{\kappa_{2}}^{3}\mu_{0}\Omega_{2}{\eta_{2}}^{2}-24448\;{\kappa_{1}}^{4}\mu_{0}\Omega_{2}), \end{array}$$
$$\begin{array}{c} \epsilon =\frac{5760}{\Omega_{1}^{4}}(-24448\;{\kappa_{1}}^{4}\mu_{0}\Omega_{2}-24448\;{\kappa_{2}}^{4}\mu_{0}\Omega_{2}+37248\;\kappa_{1}{\kappa_{2}}^{3}{\mu_{1}}^{2}\eta_{2}-50240\;\kappa_{1}{\kappa_{2}}^{3}{\mu_{1}}^{2}{\eta_{2}}^{2}\\ +1024\;\kappa_{1}{\kappa_{2}}^{3}{\mu_{1}}^{2}{\eta_{2}}^{3}-32768\;\kappa_{2}{\kappa_{1}}^{3}{\eta_{2}}^{2}{\mu_{0}}^{2}+16384\;\kappa_{2}{\kappa_{1}}^{3}{\mu_{0}}^{2}\eta_{2}+37248\;\kappa_{2}{\kappa_{1}}^{3}{\mu_{1}}^{2}\eta_{2}\\ -50240\;\kappa_{2}{\kappa_{1}}^{3}{\mu_{1}}^{2}{\eta_{2}}^{2}+1024\;\kappa_{2}{\kappa_{1}}^{3}{\mu_{1}}^{2}{\eta_{2}}^{3}-37248\;{\kappa_{2}}^{2}{\kappa_{1}}^{2}{\mu_{1}}^{2}\eta_{2}+50240\;{\kappa_{2}}^{2}{\kappa_{1}}^{2}{\mu_{1}}^{2}{\eta_{2}}^{2}\\ -1024\;{\kappa_{2}}^{2}{\kappa_{1}}^{2}{\mu_{1}}^{2}{\eta_{2}}^{3}-16384\;{\kappa_{2}}^{2}{\kappa_{1}}^{2}{\mu_{0}}^{2}\eta_{2}+32768\;{\kappa_{2}}^{2}{\kappa_{1}}^{2}{\mu_{0}}^{2}{\eta_{2}}^{2}+16384\;\kappa_{1}{\kappa_{2}}^{3}{\mu_{0}}^{2}\eta_{2}\\ -32768\;\kappa_{1}{\kappa_{2}}^{3}{\mu_{0}}^{2}{\eta_{2}}^{2}-2048\;\kappa_{2}{\kappa_{1}}^{3}{\eta_{2}}^{2}\mu_{0}\Omega_{2}+1024\;\kappa_{2}{\kappa_{1}}^{3}\mu_{0}\Omega_{2}\eta_{2}-1024\;{\kappa_{2}}^{2}{\kappa_{1}}^{2}\mu_{0}\Omega_{2}\eta_{2}\\ +2048\;{\kappa_{2}}^{2}{\kappa_{1}}^{2}\mu_{0}\Omega_{2}{\eta_{2}}^{2}+1024\;\kappa_{1}{\kappa_{2}}^{3}\mu_{0}\Omega_{2}\eta_{2}-1024\;{\kappa_{2}}^{4}\mu_{0}\Omega_{2}\eta_{2}+2048\;{\kappa_{2}}^{4}\mu_{0}\Omega_{2}{\eta_{2}}^{2}\\ -1024\;{\kappa_{1}}^{4}\mu_{0}\Omega_{2}\eta_{2}+2048\;{\kappa_{1}}^{4}\mu_{0}\Omega_{2}{\eta_{2}}^{2}+24448\;\kappa_{2}{\kappa_{1}}^{3}\mu_{0}\Omega_{2}-24448\;{\kappa_{2}}^{2}{\kappa_{1}}^{2}\mu_{0}\Omega_{2}\\ +24448\;\kappa_{1}{\kappa_{2}}^{3}\mu_{0}\Omega_{2}-6957\;\kappa_{1}{\kappa_{2}}^{3}{\mu_{1}}^{2}+391168\;\kappa_{2}{\kappa_{1}}^{3}{\mu_{0}}^{2}-6957\;\kappa_{2}{\kappa_{1}}^{3}{\mu_{1}}^{2}\\ +6957\;{\kappa_{2}}^{2}{\kappa_{1}}^{2}{\mu_{1}}^{2}-391168\;{\kappa_{2}}^{2}{\kappa_{1}}^{2}{\mu_{0}}^{2}+391168\;\kappa_{1}{\kappa_{2}}^{3}{\mu_{0}}^{2}-16384\;{\kappa_{1}}^{4}{\mu_{0}}^{2}\eta_{2}\\ +32768\;{\kappa_{1}}^{4}{\mu_{0}}^{2}{\eta_{2}}^{2}-37248\;{\kappa_{1}}^{4}{\mu_{1}}^{2}\eta_{2}+50240\;{\kappa_{1}}^{4}{\mu_{1}}^{2}{\eta_{2}}^{2}-1024\;{\kappa_{1}}^{4}{\mu_{1}}^{2}{\eta_{2}}^{3}\\ -16384\;{\kappa_{2}}^{4}{\mu_{0}}^{2}\eta_{2}+32768\;{\kappa_{2}}^{4}{\mu_{0}}^{2}{\eta_{2}}^{2}-37248\;{\kappa_{2}}^{4}{\mu_{1}}^{2}\eta_{2}+50240\;{\kappa_{2}}^{4}{\mu_{1}}^{2}{\eta_{2}}^{2}\\ -1024\;{\kappa_{2}}^{4}{\mu_{1}}^{2}{\eta_{2}}^{3}+6957\;{\kappa_{2}}^{4}{\mu_{1}}^{2}+6957\;{\kappa_{1}}^{4}{\mu_{1}}^{2}-391168\;{\kappa_{2}}^{4}{\mu_{0}}^{2}\\ -2048\;\kappa_{1}{\kappa_{2}}^{3}\mu_{0}\Omega_{2}{\eta_{2}}^{2}-391168\;{\kappa_{1}}^{4}{\mu_{0}}^{2}){\left(16\;\mu_{0}+\Omega_{2}\right)}^{2} \end{array}$$
$$\begin{array}{c} \omega_{1}=\frac{1}{4{\Omega_{1}}^{4}}\;(496166400\;{\kappa_{1}}^{5}{\mu_{1}}^{2}\mu_{0}\Omega_{2}{\eta_{2}}^{4}-9313320960\;{\kappa_{1}}^{5}{\mu_{1}}^{2}\mu_{0}\Omega_{2}{\eta_{2}}^{3}\\ +67108864\;{\kappa_{2}}^{5}{\mu_{1}}^{2}\mu_{0}\Omega_{2}{\eta_{2}}^{6}+81313376256\;{\kappa_{1}}^{5}{\mu_{1}}^{2}\mu_{0}\Omega_{2}\eta_{2}-105809080320\;{\kappa_{1}}^{5}{\mu_{1}}^{2}\mu_{0}\Omega_{2}{\eta_{2}}^{2}\\ -339738624\;{\kappa_{2}}^{5}{\mu_{1}}^{2}\mu_{0}\Omega_{2}{\eta_{2}}^{5}+67108864\;{\kappa_{1}}^{5}{\mu_{1}}^{2}\mu_{0}\Omega_{2}{\eta_{2}}^{6}-9313320960\;{\kappa_{2}}^{5}{\mu_{1}}^{2}\mu_{0}\Omega_{2}{\eta_{2}}^{3}\\ +9496166400\;{\kappa_{2}}^{5}{\mu_{1}}^{2}\mu_{0}\Omega_{2}{\eta_{2}}^{4}+81313376256\;{\kappa_{2}}^{5}{\mu_{1}}^{2}\mu_{0}\Omega_{2}\eta_{2}-105809080320\;{\kappa_{2}}^{5}{\mu_{1}}^{2}\mu_{0}\Omega_{2}{\eta_{2}}^{2}\\ +72213331968\;{\kappa_{1}}^{5}{\mu_{0}}^{3}\Omega_{2}\eta_{2}+13770946510848\;{\kappa_{1}}^{5}{\mu_{0}}^{4}+13770946510848\;{\kappa_{2}}^{5}{\mu_{0}}^{4}+559872\;\omega_{2}\eta_{2}\\ -5225472\;\omega_{2}{\eta_{2}}^{2}-92897280\;\omega_{2}{\eta_{2}}^{4}+198180864\;\omega_{2}{\eta_{2}}^{5}-264241152\;\omega_{2}{\eta_{2}}^{6}\\ +201326592\;\omega_{2}{\eta_{2}}^{7}-67108864\;\omega_{2}{\eta_{2}}^{8}+2177966961\;{\kappa_{2}}^{5}{\mu_{1}}^{4}+2177966961\;{\kappa_{1}}^{5}{\mu_{1}}^{4}\\ +27869184\;\omega_{2}{\eta_{2}}^{3}+2571730255872\;{\kappa_{1}}^{5}{\mu_{1}}^{2}{\mu_{0}}^{2}\eta_{2}-3265734574080\;{\kappa_{1}}^{5}{\mu_{1}}^{2}{\mu_{0}}^{2}{\eta_{2}}^{2}\\ -506682408960\;{\kappa_{1}}^{5}{\mu_{1}}^{2}{\mu_{0}}^{2}{\eta_{2}}^{3}+428825640960\;{\kappa_{1}}^{5}{\mu_{1}}^{2}{\mu_{0}}^{2}{\eta_{2}}^{4}+13287555072\;{\kappa_{1}}^{5}{\mu_{1}}^{2}{\mu_{0}}^{2}{\eta_{2}}^{5}\\ -9932111872\;{\kappa_{1}}^{5}{\mu_{1}}^{2}{\mu_{0}}^{2}{\eta_{2}}^{6}+5502926848\;{\kappa_{1}}^{5}{\mu_{0}}^{3}\Omega_{2}{\eta_{2}}^{4}-486986452992\;{\kappa_{1}}^{5}{\mu_{1}}^{2}{\mu_{0}}^{2}\\ -486986452992\;{\kappa_{2}}^{5}{\mu_{1}}^{2}{\mu_{0}}^{2}+1155413311488\;{\kappa_{1}}^{5}{\mu_{0}}^{4}\eta_{2}-2290291310592\;{\kappa_{1}}^{5}{\mu_{0}}^{4}{\eta_{2}}^{2}\\ -83751862272\;{\kappa_{1}}^{5}{\mu_{0}}^{4}{\eta_{2}}^{3}+88046829568\;{\kappa_{1}}^{5}{\mu_{0}}^{4}{\eta_{2}}^{4}+1155413311488\;{\kappa_{2}}^{5}{\mu_{0}}^{4}\eta_{2}\\ -2290291310592\;{\kappa_{2}}^{5}{\mu_{0}}^{4}{\eta_{2}}^{2}-83751862272\;{\kappa_{2}}^{5}{\mu_{0}}^{4}{\eta_{2}}^{3}+88046829568\;{\kappa_{2}}^{5}{\mu_{0}}^{4}{\eta_{2}}^{4}\\ -169033844736\;{\kappa_{2}}^{5}{\mu_{1}}^{4}{\eta_{2}}^{3}+116922470400\;{\kappa_{2}}^{5}{\mu_{1}}^{4}{\eta_{2}}^{4}-4431937536\;{\kappa_{2}}^{5}{\mu_{1}}^{4}{\eta_{2}}^{5}\\ -217055232\;{\kappa_{2}}^{5}{\mu_{1}}^{4}{\eta_{2}}^{6}+201326592\;{\kappa_{2}}^{5}{\mu_{1}}^{4}{\eta_{2}}^{7}-67108864\;{\kappa_{2}}^{5}{\mu_{1}}^{4}{\eta_{2}}^{8}\\ -169033844736\;{\kappa_{1}}^{5}{\mu_{1}}^{4}{\eta_{2}}^{3}+116922470400\;{\kappa_{1}}^{5}{\mu_{1}}^{4}{\eta_{2}}^{4}-4431937536\;{\kappa_{1}}^{5}{\mu_{1}}^{4}{\eta_{2}}^{5}\\ -217055232\;{\kappa_{1}}^{5}{\mu_{1}}^{4}{\eta_{2}}^{6}+201326592\;{\kappa_{1}}^{5}{\mu_{1}}^{4}{\eta_{2}}^{7}-67108864\;{\kappa_{1}}^{5}{\mu_{1}}^{4}{\eta_{2}}^{8}\\ +93885153408\;{\mu_{1}}^{4}{\eta_{2}}^{2}{\kappa_{2}}^{5}+93885153408\;{\kappa_{1}}^{5}{\mu_{1}}^{4}{\eta_{2}}^{2}-23321530368\;\eta_{2}{\kappa_{2}}^{5}{\mu_{1}}^{4}\\ -23321530368\;\eta_{2}{\kappa_{1}}^{5}{\mu_{1}}^{4}-5234491392\;{\kappa_{1}}^{5}{\mu_{0}}^{3}\Omega_{2}{\eta_{2}}^{3}+13287555072\;{\kappa_{2}}^{5}{\mu_{1}}^{2}{\mu_{0}}^{2}{\eta_{2}}^{5}\\ +428825640960\;{\kappa_{2}}^{5}{\mu_{1}}^{2}{\mu_{0}}^{2}{\eta_{2}}^{4}-506682408960\;{\kappa_{2}}^{5}{\mu_{1}}^{2}{\mu_{0}}^{2}{\eta_{2}}^{3}-15307439616\;{\kappa_{1}}^{5}{\mu_{1}}^{2}\mu_{0}\Omega_{2}\\ -3265734574080\;{\kappa_{2}}^{5}{\mu_{1}}^{2}{\mu_{0}}^{2}{\eta_{2}}^{2}+2571730255872\;{\kappa_{2}}^{5}{\mu_{1}}^{2}{\mu_{0}}^{2}\eta_{2}+860684156928\;{\kappa_{1}}^{5}{\mu_{0}}^{3}\Omega_{2}-26244\;\omega_{2}\\ +5502926848\;{\kappa_{2}}^{5}{\mu_{0}}^{3}\Omega_{2}{\eta_{2}}^{4}-9932111872\;{\kappa_{2}}^{5}{\mu_{1}}^{2}{\mu_{0}}^{2}{\eta_{2}}^{6}+860684156928\;{\kappa_{2}}^{5}{\mu_{0}}^{3}\Omega_{2}\\ +72213331968\;{\kappa_{2}}^{5}{\mu_{0}}^{3}\Omega_{2}\eta_{2}-15307439616\;{\kappa_{2}}^{5}{\mu_{1}}^{2}\Omega_{2}-143143206912\;{\kappa_{2}}^{5}{\mu_{0}}^{3}\Omega_{2}{\eta_{2}}^{2}\\ -339738624\;{\kappa_{1}}^{5}{\mu_{1}}^{2}\mu_{0}\Omega_{2}{\eta_{2}}^{5}-5234491392\;{\kappa_{2}}^{5}{\mu_{0}}^{3}\Omega_{2}{\eta_{2}}^{3}-143143206912\;{\kappa_{1}}^{5}{\mu_{0}}^{3}\Omega_{2}{\eta_{2}}^{2}), \end{array}$$
where
(41)$$\Omega_{1}=9-48\;\eta_{2}+64\;{\eta_{2}}^{2},\;\Omega_{2}=\sqrt{256\;{\mu_{0}}^{2}-9\;{\mu_{1}}^{2}+48\;{\mu_{1}}^{2}\eta_{2}-64\;{\mu_{1}}^{2}{\eta_{2}}^{2}}$$
and $\eta_{2},\mu_{0},\mu_{1},\kappa_{1},\kappa_{2}$ and $\omega_{2}$ are free parameters.

Then, the solution of Equation (Equation 1) has the following form:(42)$$\begin{array}{c} u\left(\xi_{1},\xi_{2}\right)=3+\frac{1}{8}\;\frac{\mu_{1}\Omega_{1}}{16\;\mu_{0}+\Omega_{2}}f_{1}\left(\xi_{1}\right)+f_{1}{\left(\xi_{1}\right)}^{2}\\ +\left(\frac{1}{2}\;\eta_{2}-\frac{1}{8}\right)f_{2}\left(\xi_{2}\right)+\eta_{2}f_{2}{\left(\xi_{2}\right)}^{2}+f_{2}{\left(\xi_{2}\right)}^{3}+f_{2}{\left(\xi_{2}\right)}^{4} \end{array}$$

For this case, the first simple equation is an ODE of Riccati kind:(43)$$\frac{df_{1}}{d\xi_{1}}=\mu_{0}+\mu_{1}f_{1}+8\;\frac{16\;\mu_{0}+\Omega_{2}}{\Omega_{1}}f_{1}^{2},$$
while the second simple equation is the Abel ordinary differential equation of first kind:(44)$$\begin{array}{c} \frac{df_{2}}{d\xi_{2}}=\frac{\left(\Omega_{2}+\mu_{0}\right)\left(-5+16\;\eta_{2}\right)}{16\Omega_{1}}+\frac{1}{4}\;\frac{\left(16\;\eta_{2}-3\right)\left(16\;\mu_{0}+\Omega_{2}\right)}{\Omega_{1}}f_{2}\\ +3\;\frac{16\;\mu_{0}+\Omega_{2}}{\Omega_{1}}f_{2}^{2}+4\;\frac{16\;\mu_{0}+\Omega_{2}}{\Omega_{1}}f_{2}^{3} \end{array}$$

The solutions of Equations (Equation 43) and (Equation 44) can be expressed by the special functions *V* as follows:(45)$$\begin{array}{c} f_{1}=V_{\mu_{0},\mu_{1},8\;\frac{16\;\mu_{0}+\Omega_{2}}{\Omega_{1}}}\left(\xi_{1};1,1,2\right),\\ f_{2}=V_{\frac{\left(\Omega_{2}+\mu_{0}\right)\left(-5+16\;\eta_{2}\right)}{16\Omega_{1}},\frac{1}{4}\;\frac{\left(16\;\eta_{2}-3\right)\left(16\;\mu_{0}+\Omega_{2}\right)}{\Omega_{1}},3\;\frac{16\;\mu_{0}+\Omega_{2}}{\Omega_{1}},4\;\frac{16\;\mu_{0}+\Omega_{2}}{\Omega_{1}}}\left(\xi_{2};1,1,3\right) \end{array}$$

In this way, the solution of Equation (Equation 42) is reduced to:(46)$$\begin{array}{c} u\left(\xi_{1},\xi_{2}\right)=3+\frac{1}{8}\;\frac{\mu_{1}\Omega_{1}}{16\;\mu_{0}+\Omega_{2}}V_{\mu_{0},\mu_{1},8\;\frac{16\;\mu_{0}+\Omega_{2}}{\Omega_{1}}}\left(\xi_{1};1,1,2\right)+V_{\mu_{0},\mu_{1},8\;\frac{16\;\mu_{0}+\Omega_{2}}{\Omega_{1}}}^{2}\left(\xi_{1};1,1,2\right)\\ +\left(\frac{1}{2}\;\eta_{2}-\frac{1}{8}\right)V_{\frac{\left(\Omega_{2}+\mu_{0}\right)\left(-5+16\;\eta_{2}\right)}{16\Omega_{1}},\frac{1}{4}\;\frac{\left(16\;\eta_{2}-3\right)\left(16\;\mu_{0}+\Omega_{2}\right)}{\Omega_{1}},3\;\frac{16\;\mu_{0}+\Omega_{2}}{\Omega_{1}},4\;\frac{16\;\mu_{0}+\Omega_{2}}{\Omega_{1}}}\left(\xi_{2};1,1,3\right)\\ +\eta_{2}V_{\frac{\left(\Omega_{2}+\mu_{0}\right)\left(-5+16\;\eta_{2}\right)}{16\Omega_{1}},\frac{1}{4}\;\frac{\left(16\;\eta_{2}-3\right)\left(16\;\mu_{0}+\Omega_{2}\right)}{\Omega_{1}},3\;\frac{16\;\mu_{0}+\Omega_{2}}{\Omega_{1}},4\;\frac{16\;\mu_{0}+\Omega_{2}}{\Omega_{1}}}^{2}\left(\xi_{2};1,1,3\right)\\ +V_{\frac{\left(\Omega_{2}+\mu_{0}\right)\left(-5+16\;\eta_{2}\right)}{16\Omega_{1}},\frac{1}{4}\;\frac{\left(16\;\eta_{2}-3\right)\left(16\;\mu_{0}+\Omega_{2}\right)}{\Omega_{1}},3\;\frac{16\;\mu_{0}+\Omega_{2}}{\Omega_{1}},4\;\frac{16\;\mu_{0}+\Omega_{2}}{\Omega_{1}}}^{3}\left(\xi_{2},1,1,3\right)\\ +V_{\frac{\left(\Omega_{2}+\mu_{0}\right)\left(-5+16\;\eta_{2}\right)}{16\Omega_{1}},\frac{1}{4}\;\frac{\left(16\;\eta_{2}-3\right)\left(16\;\mu_{0}+\Omega_{2}\right)}{\Omega_{1}},3\;\frac{16\;\mu_{0}+\Omega_{2}}{\Omega_{1}},4\;\frac{16\;\mu_{0}+\Omega_{2}}{\Omega_{1}}}^{4}\left(\xi_{2},1,1,3\right) \end{array}$$In the context of the general solution of the Riccati ordinary differential equation (see Equation (Equation 17)), the special function $V_{\mu_{0},\mu_{1},8\;\frac{16\;\mu_{0}+\Omega_{2}}{\Omega_{1}}}\left(\xi_{1},1,1,2\right)$ reduces to the following specific forms:(47)$$\begin{array}{c} V_{\mu_{0},\mu_{1},8\;\frac{16\;\mu_{0}+\Omega_{2}}{\Omega_{1}}}\left(\xi_{1};1,1,2\right)=-\frac{1}{16}\frac{\mu_{1}\Omega_{1}}{\left(16\;\mu_{0}+\Omega_{2}\right)}-\frac{1}{16}\frac{\Omega_{3}}{\left(16\;\mu_{0}+\Omega_{2}\right)}\tanh\left(\frac{1}{2}\Omega_{3}\xi_{1}\right)\\ +\frac{1}{2}\frac{\exp\left(\frac{1}{2}\Omega_{3}\xi_{1}\right)}{\cosh\left(\frac{1}{2}\Omega_{3}\xi_{1}\right)\left(\frac{\mu_{2}\sqrt{\Omega_{1}}}{\Omega_{3}}+2C\exp\left(\frac{1}{2}\Omega_{3}\xi_{1}\right)\cosh\left(\frac{1}{2}\Omega_{3}\xi_{1}\right)\right)} \end{array}$$
where
(48)$$\Omega_{3}=\sqrt{\frac{\mu_{1}^{2}\Omega_{1}-32\mu_{0}\left(\;16\;\mu_{0}+\Omega_{2}\right)}{\Omega_{1}}}>0$$
and $\xi_{1}=\kappa_{1}x+\omega_{1}t$, as the expression for $\omega_{1}$ is given in Equation (Equation 40) and $\kappa_{1}$ is a free parameter. In Equation (Equation 47), *C* is a constant of integration.

For the special case $\frac{\left(\Omega_{2}+\mu_{0}\right)\left(-5+16\;\eta_{2}\right)}{16\Omega_{1}}=\frac{1}{4}\left(\frac{\left(16\;\eta_{2}-3\right)\left(16\;\mu_{0}+\Omega_{2}\right)}{\Omega_{1}}-\frac{1}{6}\frac{16\;\mu_{0}+\Omega_{2}}{\Omega_{1}}\right)$, the special function $V_{\frac{\left(\Omega_{2}+\mu_{0}\right)\left(-5+16\;\eta_{2}\right)}{16\Omega_{1}},\frac{1}{4}\;\frac{\left(16\;\eta_{2}-3\right)\left(16\;\mu_{0}+\Omega_{2}\right)}{\Omega_{1}},3\;\frac{16\;\mu_{0}+\Omega_{2}}{\Omega_{1}},4\;\frac{16\;\mu_{0}+\Omega_{2}}{\Omega_{1}}}\left(\xi_{2};1,1,3\right)$ can be presented by the solution of the Abel ordinary differential equation of first kind (see Equation (Equation 14)):(49)$$\begin{array}{c} V_{\frac{\left(\Omega_{2}+\mu_{0}\right)\left(-5+16\;\eta_{2}\right)}{16\Omega_{1}},\frac{1}{4}\;\frac{\left(16\;\eta_{2}-3\right)\left(16\;\mu_{0}+\Omega_{2}\right)}{\Omega_{1}},3\;\frac{16\;\mu_{0}+\Omega_{2}}{\Omega_{1}},4\;\frac{16\;\mu_{0}+\Omega_{2}}{\Omega_{1}}}\left(\xi_{2};1,1,3\right)\\ =\frac{\exp\left[\left(\frac{\left(16\;\eta_{2}-3\right)\left(16\;\mu_{0}+\Omega_{2}\right)}{\Omega_{1}}-\frac{1}{6}\frac{16\;\mu_{0}+\Omega_{2}}{\Omega_{1}}\right)\xi_{2}\right]}{\sqrt{C^{*}-3\;\frac{16\;\mu_{0}+\Omega_{2}}{\Omega_{1}}}\exp\left[\left(\frac{\left(16\;\eta_{2}-3\right)\left(16\;\mu_{0}+\Omega_{2}\right)}{\Omega_{1}}-\frac{1}{6}\frac{16\;\mu_{0}+\Omega_{2}}{\Omega_{1}}\right)\xi_{2}\right]}-\frac{1}{4}, \end{array}$$
where $\xi_{2}=\kappa_{2}x+\omega_{2}t$, as $\kappa_{2}$ and $\omega_{2}$ are free parameters. In Equation (Equation 49), $C^{*}$ is a constant of integration. Then, the solution of Equation (Equation 1) presented in its initial coordinates is
(50)$$\begin{array}{c} u\left(x,t\right)=3-\frac{1}{8}\;\frac{\mu_{1}\Omega_{1}}{16\;\mu_{0}+\Omega_{2}}[\frac{1}{16}\frac{\mu_{1}\Omega_{1}}{\left(16\;\mu_{0}+\Omega_{2}\right)}+\frac{1}{16}\frac{\Omega_{3}}{\left(16\;\mu_{0}+\Omega_{2}\right)}\tanh\left[\frac{1}{2}\Omega_{3}\left(\kappa_{1}x+\omega_{1}t\right)\right]\\ -\frac{1}{2}\frac{\exp\left[\frac{1}{2}\Omega_{3}\left(\kappa_{1}x+\omega_{1}t\right)\right]}{\cosh\left[\frac{1}{2}\Omega_{3}\left(\kappa_{1}x+\omega_{1}t\right)\right]\left(\frac{\mu_{2}\sqrt{\Omega_{1}}}{\Omega_{3}}+2C\exp\left[\frac{1}{2}\Omega_{3}\left(\kappa_{1}x+\omega_{1}t\right)\right]\cosh\left[\frac{1}{2}\Omega_{3}\left(\kappa_{1}x+\omega_{1}t\right)\right]\right)}]\\ -[\frac{1}{16}\frac{\mu_{1}\Omega_{1}}{\left(16\;\mu_{0}+\Omega_{2}\right)}+\frac{1}{16}\frac{\Omega_{3}}{\left(16\;\mu_{0}+\Omega_{2}\right)}\tanh\left[\frac{1}{2}\Omega_{3}\left(\kappa_{1}x+\omega_{1}t\right)\right]\\ -\frac{1}{2}\frac{\exp\left[\frac{1}{2}\Omega_{3}\left(\kappa_{1}x+\omega_{1}t\right)\right]}{\cosh\left[\frac{1}{2}\Omega_{3}\left(\kappa_{1}x+\omega_{1}t\right)\right]\left(\frac{\mu_{2}\sqrt{\Omega_{1}}}{\Omega_{3}}+2C\exp\left[\frac{1}{2}\Omega_{3}\left(\kappa_{1}x+\omega_{1}t\right)\right]\cosh\left[\frac{1}{2}\Omega_{3}\left(\kappa_{1}x+\omega_{1}t\right)\right]\right)}{]}^{2}\\ +\left(\frac{1}{2}\;\eta_{2}-\frac{1}{8}\right)\left[\frac{\exp\left[\left(\frac{\left(16\;\eta_{2}-3\right)\left(16\;\mu_{0}+\Omega_{2}\right)}{\Omega_{1}}-\frac{1}{6}\frac{16\;\mu_{0}+\Omega_{2}}{\Omega_{1}}\right)\left(\kappa_{2}x+\omega_{2}t\right)\right]}{\sqrt{C^{*}-3\;\frac{16\;\mu_{0}+\Omega_{2}}{\Omega_{1}}}\exp\left[\left(\frac{\left(32\;\eta_{2}-6\right)\left(16\;\mu_{0}+\Omega_{2}\right)}{\Omega_{1}}-\frac{1}{6}\frac{16\;\mu_{0}+\Omega_{2}}{\Omega_{1}}\right)\left(\kappa_{2}t+\omega_{2}t\right)\right]}-\frac{1}{4}\right]\\ +\eta_{2}{\left[\frac{\exp\left[\left(\frac{\left(16\;\eta_{2}-3\right)\left(16\;\mu_{0}+\Omega_{2}\right)}{\Omega_{1}}-\frac{1}{6}\frac{16\;\mu_{0}+\Omega_{2}}{\Omega_{1}}\right)\left(\kappa_{2}x+\omega_{2}t\right)\right]}{\sqrt{C^{*}-3\;\frac{16\;\mu_{0}+\Omega_{2}}{\Omega_{1}}}\exp\left[\left(\frac{\left(32\;\eta_{2}-6\right)\left(16\;\mu_{0}+\Omega_{2}\right)}{\Omega_{1}}-\frac{1}{6}\frac{16\;\mu_{0}+\Omega_{2}}{\Omega_{1}}\right)\left(\kappa_{2}t+\omega_{2}t\right)\right]}-\frac{1}{4}\right]}^{2}\\ +{\left[\frac{\exp\left[\left(\frac{\left(16\;\eta_{2}-3\right)\left(16\;\mu_{0}+\Omega_{2}\right)}{\Omega_{1}}-\frac{1}{6}\frac{16\;\mu_{0}+\Omega_{2}}{\Omega_{1}}\right)\left(\kappa_{2}x+\omega_{2}t\right)\right]}{\sqrt{C^{*}-3\;\frac{16\;\mu_{0}+\Omega_{2}}{\Omega_{1}}}\exp\left[\left(\frac{\left(32\;\eta_{2}-6\right)\left(16\;\mu_{0}+\Omega_{2}\right)}{\Omega_{1}}-\frac{1}{6}\frac{16\;\mu_{0}+\Omega_{2}}{\Omega_{1}}\right)\left(\kappa_{2}t+\omega_{2}t\right)\right]}-\frac{1}{4}\right]}^{3}\\ +{\left[\frac{\exp\left[\left(\frac{\left(16\;\eta_{2}-3\right)\left(16\;\mu_{0}+\Omega_{2}\right)}{\Omega_{1}}-\frac{1}{6}\frac{16\;\mu_{0}+\Omega_{2}}{\Omega_{1}}\right)\left(\kappa_{2}x+\omega_{2}t\right)\right]}{\sqrt{C^{*}-3\;\frac{16\;\mu_{0}+\Omega_{2}}{\Omega_{1}}}\exp\left[\left(\frac{\left(32\;\eta_{2}-6\right)\left(16\;\mu_{0}+\Omega_{2}\right)}{\Omega_{1}}-\frac{1}{6}\frac{16\;\mu_{0}+\Omega_{2}}{\Omega_{1}}\right)\left(\kappa_{2}t+\omega_{2}t\right)\right]}-\frac{1}{4}\right]}^{4} \end{array}$$

For the particular case $\nu_{0}=\nu_{2}=0$, one non-trivial solution of the system (Equation 27) has the following form: (51)$$\begin{array}{c} \zeta_{0}=\zeta_{2}=1,\;\zeta_{1}=-\frac{3}{8}\;\frac{\mu_{1}}{\nu_{1}},\;\eta_{0}=\eta_{3}=\eta_{4}=1,\;\eta_{1}=\frac{11}{32},\;\eta_{2}=\frac{15}{16},\;\nu_{0}=\nu_{2}=0,\\ \nu_{3}=-\frac{4}{3}\nu_{1},\;\mu_{0}=-\frac{3}{128}\;\frac{4\;{\mu_{1}}^{2}-9\;{\nu_{1}}^{2}}{\nu_{1}},\;\mu_{2}=-\frac{8}{3}\nu_{1},\;\alpha =\frac{640}{3}\;\frac{\left({\kappa_{1}}^{5}+{\kappa_{2}}^{5}\right){\nu_{1}}^{2}}{{\kappa_{2}}^{4}+{\kappa_{1}}^{4}}\\ \beta =-\frac{640}{3}\;\frac{\left({\kappa_{1}}^{4}-\kappa_{2}{\kappa_{1}}^{3}+{\kappa_{2}}^{2}{\kappa_{1}}^{2}-{\kappa_{2}}^{3}\kappa_{1}+{\kappa_{2}}^{4}\right){\nu_{1}}^{2}}{{\kappa_{1}}^{2}-\kappa_{2}\kappa_{1}+{\kappa_{2}}^{2}}\\ \gamma =\frac{81920}{9}\;\left({\kappa_{1}}^{4}-\kappa_{2}{\kappa_{1}}^{3}+{\kappa_{2}}^{2}{\kappa_{1}}^{2}-{\kappa_{2}}^{3}\kappa_{1}+{\kappa_{2}}^{4}\right){\nu_{1}}^{4}\\ \epsilon =-\frac{32}{9}\;{\nu_{1}}^{2}(-2485\;{\kappa_{1}}^{4}{\nu_{1}}^{2}+2485\;\kappa_{2}{\kappa_{1}}^{3}{\nu_{1}}^{2}+24\;{\kappa_{1}}^{2}\delta \\ -2485\;{\kappa_{2}}^{2}{\kappa_{1}}^{2}{\nu_{1}}^{2}-24\;\kappa_{2}\kappa_{1}\delta +2485\;{\kappa_{2}}^{3}\kappa_{1}{\nu_{1}}^{2}+24\;{\kappa_{2}}^{2}\delta -2485\;{\kappa_{2}}^{4}{\nu_{1}}^{2})\\ \omega_{1}=\frac{1}{14400}\frac{1}{\kappa_{1}^{4}-\kappa_{2}\kappa_{1}^{3}+\kappa_{2}^{2}\kappa_{1}^{2}-\kappa_{2}^{3}\kappa_{1}+\kappa_{2}^{4}}(-2160\;\kappa_{1}{\kappa_{2}}^{6}\delta \;{\mu_{1}}^{2}-2160\;{\kappa_{1}}^{6}\delta \;{\mu_{1}}^{2}\kappa_{2}\\ +2160\;\kappa_{1}^{5}\delta \;\mu_{1}^{2}\kappa_{2}^{2}+2160\;\kappa_{1}^{2}\delta \;\mu_{1}^{2}\kappa_{2}^{5}+806520\;\kappa_{1}\kappa_{2}^{6}\delta \;\nu_{1}^{2}+1575900\;\kappa_{1}{\kappa_{2}}^{8}{\mu_{1}}^{2}{\nu_{1}}^{2}\\ +1575900\;{\kappa_{1}}^{8}{\mu_{1}}^{2}\kappa_{2}{\nu_{1}}^{2}-1575900\;{\kappa_{1}}^{7}{\mu_{1}}^{2}{\kappa_{2}}^{2}{\nu_{1}}^{2}+1575900\;{\kappa_{1}}^{6}{\mu_{1}}^{2}{\kappa_{2}}^{3}{\nu_{1}}^{2}\\ +806520\;{\kappa_{1}}^{6}{\nu_{1}}^{2}\kappa_{2}\delta -806520\;{\kappa_{1}}^{5}{\kappa_{2}}^{2}\delta \;{\nu_{1}}^{2}-1575900\;{\kappa_{1}}^{5}{\kappa_{2}}^{4}{\mu_{1}}^{2}{\nu_{1}}^{2}\\ -1575900\;{\kappa_{1}}^{4}{\kappa_{2}}^{5}{\mu_{1}}^{2}{\nu_{1}}^{2}+1575900\;{\kappa_{1}}^{3}{\kappa_{2}}^{6}{\mu_{1}}^{2}{\nu_{1}}^{2}-806520\;{\kappa_{1}}^{2}{\kappa_{2}}^{5}\delta \;{\nu_{1}}^{2}\\ -1575900\;{\kappa_{1}}^{2}{\kappa_{2}}^{7}{\mu_{1}}^{2}{\nu_{1}}^{2}+2160\;{\kappa_{1}}^{7}\delta \;{\mu_{1}}^{2}+106965025\;{\kappa_{1}}^{7}{\nu_{1}}^{4}{\kappa_{2}}^{2}\\ -806520\;{\kappa_{1}}^{7}{\nu_{1}}^{2}\delta -8100\;{\kappa_{1}}^{8}{\mu_{1}}^{4}\kappa_{2}+3024\;{\kappa_{1}}^{3}{\kappa_{2}}^{2}\delta^{2}+8100\;{\kappa_{1}}^{7}{\mu_{1}}^{4}{\kappa_{2}}^{2}\\ +14400\;{\kappa_{1}}^{3}\omega_{2}\kappa_{2}+2160\;\delta \;{\mu_{1}}^{2}{\kappa_{2}}^{7}+8100\;{\kappa_{1}}^{4}{\kappa_{2}}^{5}{\mu_{1}}^{4}+8100\;{\kappa_{1}}^{5}\kappa_{2}^{4}{\mu_{1}}^{4}\\ -8100\;{\kappa_{1}}^{6}{\mu_{1}}^{4}{\kappa_{2}}^{3}-8100\;{\kappa_{1}}^{3}{\kappa_{2}}^{6}{\mu_{1}}^{4}-3024\;\kappa_{1}^{4}\delta^{2}\kappa_{2}-14400\;{\kappa_{1}}^{2}\omega_{2}{\kappa_{2}}^{2}\\ -1575900\;{\kappa_{2}}^{9}{\mu_{1}}^{2}{\nu_{1}}^{2}-14400\;\omega_{2}{\kappa_{2}}^{4}+3024\;{\kappa_{2}}^{5}\delta^{2}+8100\;{\kappa_{2}}^{9}{\mu_{1}}^{4}\\ +106965025\;{\kappa_{2}}^{9}{\nu_{1}}^{4}+8100\;{\kappa_{1}}^{9}{\mu_{1}}^{4}+3024\;{\kappa_{1}}^{5}\delta^{2}-14400\;{\kappa_{1}}^{4}\omega_{2}\\ +106965025\;{\kappa_{1}}^{9}{\nu_{1}}^{4}-1575900\;{\kappa_{1}}^{9}{\mu_{1}}^{2}{\nu_{1}}^{2}-106965025\;{\kappa_{1}}^{6}{\nu_{1}}^{4}{\kappa_{2}}^{3}\\ +106965025\;{\kappa_{1}}^{4}{\kappa_{2}}^{5}{\nu_{1}}^{4}-106965025\;\kappa_{1}\kappa_{2}-106965025\;\kappa_{1}\kappa_{2}\\ +106965025\;{\kappa_{1}}^{5}{\kappa_{2}}^{4}{\nu_{1}}^{4}+106965025\;{\kappa_{1}}^{2}{\kappa_{2}}^{7}{\nu_{1}}^{4}-106965025\;\kappa_{1}\kappa_{2}\\ -106965025\;{\kappa_{1}}^{8}{\nu_{1}}^{4}\kappa_{2}-806520\;{\kappa_{2}}^{7}\delta \;{\nu_{1}}^{2}-106965025\;{\kappa_{1}}^{3}{\kappa_{2}}^{6}{\nu_{1}}^{4}\\ +3024\;{\kappa_{1}}^{2}{\kappa_{2}}^{3}\delta^{2}+8100\;{\kappa_{1}}^{2}{\kappa_{2}}^{7}{\mu_{1}}^{4}-106965025\;\kappa_{1}\kappa_{2}\\ -8100\;\kappa_{1}{\kappa_{2}}^{8}{\mu_{1}}^{4}+14400\;\kappa_{1}\omega_{2}{\kappa_{2}}^{3}-3024\;\kappa_{1}{\kappa_{2}}^{4}\delta^{2}) \end{array}$$
where $\delta ,\mu_{1},\nu_{1},\kappa_{1},\kappa_{2}$ and $\omega_{2}$ are free parameters.

For this case, the solution of Equation (Equation 1) can be presented as
(52)$$u\left(\xi_{1},\xi_{2}\right)=3-\frac{3}{8}\frac{\mu_{1}}{\nu_{1}}f_{1}\left(\xi_{1}\right)+f_{1}{\left(\xi_{1}\right)}^{2}+\frac{11}{32}f_{2}\left(\xi_{2}\right)+\frac{15}{16}f_{2}{\left(\xi_{2}\right)}^{2}+f_{2}{\left(\xi_{2}\right)}^{3}+f_{2}{\left(\xi_{2}\right)}^{4}$$
where the simple equations are of Riccati kind and of Bernoulli kind, as follows:(53)$$\frac{df_{1}}{d\xi_{1}}=-\frac{3}{128}\;\frac{4\;{\mu_{1}}^{2}-9\;{\nu_{1}}^{2}}{\nu_{1}}+\mu_{1}f_{1}-\frac{8}{3}\nu_{1}f_{1}^{2},\;\;\;\frac{df_{2}}{d\xi_{2}}=\nu_{1}f_{2}-\frac{4}{3}\nu_{1}f_{2}^{3}$$

We present again the solutions of Equation (Equation 53) by the special functions *V*:(54)$$f_{1}=V_{-\frac{3}{128}\;\frac{4\;{\mu_{1}}^{2}-9\;{\nu_{1}}^{2}}{\nu_{1}},\mu_{1},-\frac{8}{3}\nu_{1}}\left(\xi_{1};1,1,2\right),\;\;\;f_{2}=V_{0,\nu_{1},0,-\frac{4}{3}\nu_{1}}\left(\xi_{2};1,1,3\right)$$

Then, the solution of Equation (Equation 52) reduces to the following form: (55)$$\begin{array}{c} u\left(\xi_{1},\xi_{2}\right)=3-\frac{3}{8}\frac{\mu_{1}}{\nu_{1}}V_{-\frac{3}{128}\;\frac{4\;{\mu_{1}}^{2}-9\;{\nu_{1}}^{2}}{\nu_{1}},\mu_{1},-\frac{8}{3}\nu_{1}}\left(\xi_{1};1,1,2\right)+V_{-\frac{3}{128}\;\frac{4\;{\mu_{1}}^{2}-9\;{\nu_{1}}^{2}}{\nu_{1}},\mu_{1},-\frac{8}{3}\nu_{1}}^{2}\left(\xi_{1};1,1,2\right)\\ +\frac{11}{32}V_{0,\nu_{1},0,-\frac{4}{3}\nu_{1}}\left(\xi_{2};1,1,3\right)+\frac{15}{16}V_{0,\nu_{1},0,-\frac{4}{3}\nu_{1}}^{2}\left(\xi_{2};1,1,3\right)\\ +V_{0,\nu_{1},0,-\frac{4}{3}\nu_{1}}^{3}\left(\xi_{2};1,1,3\right)+V_{0,\nu_{1},0,-\frac{4}{3}\nu_{1}}^{4}\left(\xi_{2};1,1,3\right) \end{array}$$
where in the context of the general solution of the Riccati differential equation (see Equation (Equation 17), the special function $V_{-\frac{3}{128}\;\frac{4\;{\mu_{1}}^{2}-9\;{\nu_{1}}^{2}}{\nu_{1}},\mu_{1},-\frac{8}{3}\nu_{1}}\left(\xi_{1};1,1,2\right)$ assumes the following form:(56)$$\begin{array}{c} V_{-\frac{3}{128}\;\frac{4\;{\mu_{1}}^{2}-9\;{\nu_{1}}^{2}}{\nu_{1}},\mu_{1},-\frac{8}{3}\nu_{1}}\left(\xi_{1};1,1,2\right)=\frac{3}{16}\frac{\mu_{1}}{\nu_{1}}+\frac{9}{16}\tanh\left(\frac{3}{2}\nu_{1}\xi_{1}\right)\\ +\frac{1}{2}\frac{\exp(\frac{3}{2}\nu_{1}\xi_{1})}{\cosh\left(\frac{3}{2}\nu_{1}\xi_{1}\right)\left(-\frac{8}{9}+2C_{1}^{*}\exp\left(\frac{3}{2}\nu_{1}\xi_{1}\right)\cosh\left(\frac{3}{2}\nu_{1}\xi_{1}\right)\right)} \end{array}$$
for $\nu_{1}>0$. In Equation (Equation 56), $\xi_{1}=\kappa_{1}x+\omega_{1}t$, where the expression of $\omega_{1}$ is presented in Equation (Equation 51) and $\kappa_{1}$ is a free parameter. In addition, $C_{1}^{*}$ is a constant of integration.

On the other hand, in the context of the general solutions of the Bernoulli differential equation (see Equations (Equation 11) and (Equation 12), the special function $V_{0,\nu_{1},0,-\frac{4}{3}\nu_{1}}\left(\xi_{2};1,1,3\right)$ is presented only by the following solution:(57)$$V_{0,\nu_{1},0,-\frac{4}{3}\nu_{1}}\left(\xi_{2};1,1,3\right)=\sqrt{\frac{\nu_{1}\exp\left(2\nu_{1}\xi_{2}\right)}{1-\frac{4}{3}\nu_{1}\exp\left(2\nu_{1}\xi_{2}\right)}}$$
for $\nu_{1}>0$, because of the restriction for the solution $V_{-\frac{3}{128}\;\frac{4\;{\mu_{1}}^{2}-9\;{\nu_{1}}^{2}}{\nu_{1}},\mu_{1},-\frac{8}{3}\nu_{1}}\left(\xi_{1};1,1,2\right)$ given above. In Equation (Equation 57), $\xi_{2}=\kappa_{2}x+\omega_{2}t$, where $\kappa_{2}$ and $\omega_{2}$ are free parameters.

For this particular case, the solution of Equation (Equation 1) written in its primary coordinates, takes the form: (58)$$\begin{array}{c} u\left(x,t\right)=3-\frac{3}{8}\frac{\mu_{1}}{\nu_{1}}[\frac{3}{16}\frac{\mu_{1}}{\nu_{1}}+\frac{9}{16}\tanh\left[\frac{3}{2}\nu_{1}\left(\kappa_{1}x+\omega_{1}t\right)\right]\\ +\frac{1}{2}\frac{\exp\left[\frac{3}{2}\nu_{1}\left(\kappa_{1}x+\omega_{1}t\right)\right]}{\cosh\left[\frac{3}{2}\nu_{1}\left(\kappa_{1}x+\omega_{1}t\right)\right]\left(-\frac{8}{9}+2C_{1}^{*}\exp\left[\frac{3}{2}\nu_{1}\left(\kappa_{1}x+\omega_{1}t\right)\right]\cosh\left[\frac{3}{2}\nu_{1}\left(\kappa_{1}x+\omega_{1}t\right)\right]\right)}]\\ +[\frac{3}{16}\frac{\mu_{1}}{\nu_{1}}+\frac{9}{16}\tanh\left[\frac{3}{2}\nu_{1}\left(\kappa_{1}x+\omega_{1}t\right)\right]\\ +\frac{1}{2}\frac{\exp\left[\frac{3}{2}\nu_{1}\left(\kappa_{1}x+\omega_{1}t\right)\right]}{\cosh\left[\frac{3}{2}\nu_{1}\left(\kappa_{1}x+\omega_{1}t\right)\right]\left(-\frac{8}{9}+2C_{1}^{*}\exp\left[\frac{3}{2}\nu_{1}\left(\kappa_{1}x+\omega_{1}t\right)\right]\cosh\left[\frac{3}{2}\nu_{1}\left(\kappa_{1}x+\omega_{1}t\right)\right]\right)}{]}^{2}\\ +\frac{11}{32}\left[\sqrt{\frac{\nu_{1}\exp\left[2\nu_{1}\left(\kappa_{2}x+\omega_{2}t\right)\right]}{1-\frac{4}{3}\nu_{1}\exp\left[2\nu_{1}\left(\kappa_{2}x+\omega_{2}t\right)\right]}}\right]+\frac{15}{16}{\left[\sqrt{\frac{\nu_{1}\exp\left[2\nu_{1}\left(\kappa_{2}x+\omega_{2}t\right)\right]}{1-\frac{4}{3}\nu_{1}\exp\left[2\nu_{1}\left(\kappa_{2}x+\omega_{2}t\right)\right]}}\right]}^{2}\\ +{\left[\sqrt{\frac{\nu_{1}\exp\left[2\nu_{1}\left(\kappa_{2}x+\omega_{2}t\right)\right]}{1-\frac{4}{3}\nu_{1}\exp\left[2\nu_{1}\left(\kappa_{2}x+\omega_{2}t\right)\right]}}\right]}^{3}+{\left[\sqrt{\frac{\nu_{1}\exp\left[2\nu_{1}\left(\kappa_{2}x+\omega_{2}t\right)\right]}{1-\frac{4}{3}\nu_{1}\exp\left[2\nu_{1}\left(\kappa_{2}x+\omega_{2}t\right)\right]}}\right]}^{4} \end{array}$$

### 3.3. Case $m_{1}=2,\;m_{2}=2$

For this case we assume that $\mu_{3}=0$ and $\nu_{3}=0$. Then, according to the balance Equation (Equation 23), $n_{1}=n_{2}=2$. The general solution of Equation (Equation 1) becomes
(59)$$u\left(\xi_{1},\xi_{2}\right)=1+\sum_{i_{1}=0}^{2}\zeta_{i_{1}}{\left[f_{1}\left(\xi_{1}\right)\right]}^{i_{1}}+\sum_{i_{2}=0}^{2}\eta_{i_{2}}{\left[f_{2}\left(\xi_{2}\right)\right]}^{i_{2}}$$
where the simple equations are of Riccati kind:(60)$$\frac{df_{1}}{d\xi_{1}}=\mu_{0}+\mu_{1}f_{1}+\mu_{2}f_{1}^{2},\;\;\frac{df_{2}}{d\xi_{2}}=\nu_{0}+\nu_{1}f_{2}+\nu_{2}f_{2}^{2}$$

One non-trivial solution of the reduced variant of the system (Equation 27) is: (61)$$\begin{array}{c} \zeta_{0}=\zeta_{1}=\zeta_{2}=1,\;\eta_{0}=\eta_{1}=\eta_{2}=1,\;\omega_{1}=-\frac{20}{9}\;{\kappa_{1}}^{5}{\mu_{2}}^{4}-\omega_{2}-\frac{20}{9}\;{\mu_{2}}^{4}{\kappa_{2}}^{5},\\ \alpha =30\;\frac{\left({\kappa_{1}}^{5}+{\kappa_{2}}^{5}\right){\mu_{2}}^{2}}{{\kappa_{2}}^{4}+{\kappa_{1}}^{4}},\;\beta =-30\;\frac{\left({\kappa_{1}}^{4}-\kappa_{2}{\kappa_{1}}^{3}+{\kappa_{2}}^{2}{\kappa_{1}}^{2}-{\kappa_{2}}^{3}\kappa_{1}+{\kappa_{2}}^{4}\right){\mu_{2}}^{2}}{{\kappa_{1}}^{2}-\kappa_{2}\kappa_{1}+{\kappa_{2}}^{2}},\\ \gamma =180\;\left({\kappa_{1}}^{4}-\kappa_{2}{\kappa_{1}}^{3}+{\kappa_{2}}^{2}{\kappa_{1}}^{2}-{\kappa_{2}}^{3}\kappa_{1}+{\kappa_{2}}^{4}\right){\mu_{2}}^{4},\;\mu_{0}=-\frac{2}{3}\;\mu_{2},\;\mu_{1}=\mu_{2},\\ \epsilon =20\;\left({\kappa_{1}}^{4}-\kappa_{2}{\kappa_{1}}^{3}+{\kappa_{2}}^{2}{\kappa_{1}}^{2}-{\kappa_{2}}^{3}\kappa_{1}+{\kappa_{2}}^{4}\right){\mu_{2}}^{4},\;\nu_{0}=-\frac{2}{3}\;\nu_{2},\\ \delta =-\frac{5}{3}\;\frac{\left({\kappa_{1}}^{4}-\kappa_{2}{\kappa_{1}}^{3}+{\kappa_{2}}^{2}{\kappa_{1}}^{2}-{\kappa_{2}}^{3}\kappa_{1}+{\kappa_{2}}^{4}\right){\mu_{2}}^{2}}{{\kappa_{1}}^{2}-\kappa_{2}\kappa_{1}+{\kappa_{2}}^{2}},\;\nu_{1}=\nu_{2} \end{array}$$
where $\mu_{2},\nu_{2},\kappa_{1},\kappa_{2}$ and $\omega_{2}$ are free parameters. Then, the solution of Equation (Equation 1) reduces to:(62)$$u\left(\xi_{1},\xi_{2}\right)=3+f_{1}\left(\xi_{1}\right)+f_{1}^{2}\left(\xi_{1}\right)+f_{2}\left(\xi_{2}\right)+f_{2}^{2}\left(\xi_{2}\right)$$
where
(63)$$\frac{df_{1}}{d\xi_{1}}=-\frac{2}{3}\mu_{2}+\mu_{2}f_{1}+\mu_{2}f_{1}^{2},\;\;\frac{df_{2}}{d\xi_{2}}=-\frac{2}{3}\nu_{2}+\nu_{2}f_{2}+\nu_{2}f_{2}^{2}$$

We present the solutions of Equation (Equation 63) by the special functions *V* as follows:(64)$$f_{1}=V_{-\frac{2}{3}\mu_{2},\mu_{2},\mu_{2}}\left(\xi_{1};1,1,2\right),\;\;f_{2}=V_{-\frac{2}{3}\nu_{2},\nu_{2},\nu_{2}}\left(\xi_{2};1,1,2\right),$$

Then Equation (Equation 62) transforms to
(65)$$\begin{array}{c} u\left(\xi_{1},\xi_{2}\right)=3+V_{-\frac{2}{3}\mu_{2},\mu_{2},\mu_{2}}\left(\xi_{1};1,1,2\right)+V_{-\frac{2}{3}\mu_{2},\mu_{2},\mu_{2}}^{2}\left(\xi_{1};1,1,2\right)\\ +V_{-\frac{2}{3}\nu_{2},\nu_{2},\nu_{2}}\left(\xi_{2};1,1,2\right)+V_{-\frac{2}{3}\nu_{2},\nu_{2},\nu_{2}}^{2}\left(\xi_{2};1,1,2\right) \end{array}$$
where
(66)$$\begin{array}{c} V_{-\frac{2}{3}\mu_{2},\mu_{2},\mu_{2}}\left(\xi_{1};1,1,2\right)=-\frac{1}{2}-\frac{\sqrt{33}}{6}\tanh\left(\frac{\sqrt{33}}{6}\mu_{2}\xi_{1}\right)\\ +\frac{1}{2}\frac{\exp(\frac{\sqrt{33}}{6}\mu_{2}\xi_{1})}{\cosh\left(\frac{\sqrt{33}}{6}\mu_{2}\xi_{1}\right)\left(\frac{\sqrt{33}}{11}+2C_{2}\exp\left(\frac{\sqrt{33}}{6}\mu_{2}\xi_{1}\right)\cosh\left(\frac{\sqrt{33}}{6}\mu_{2}\xi_{1}\right)\right)}\\ V_{-\frac{2}{3}\nu_{2},\nu_{2},\nu_{2}}\left(\xi_{2};1,1,2\right)=-\frac{1}{2}-\frac{\sqrt{33}}{6}\tanh\left(\frac{\sqrt{33}}{6}\nu_{2}\xi_{2}\right)\\ +\frac{1}{2}\frac{\exp(\frac{\sqrt{33}}{6}\nu_{2}\xi_{2})}{\cosh\left(\frac{\sqrt{33}}{6}\nu_{2}\xi_{2}\right)\left(\frac{\sqrt{33}}{11}+2C_{2}^{*}\exp\left(\frac{\sqrt{33}}{6}\nu_{2}\xi_{2}\right)\cosh\left(\frac{\sqrt{33}}{6}\nu_{2}\xi_{2}\right)\right)} \end{array}$$
for $\mu_{2}>0$ and $\nu_{2}>0$. In Equation (Equation 66), $\xi_{1}=\kappa_{1}x+\omega_{1}t$ and $\xi_{2}=\kappa_{2}x+\omega_{2}t$, where the expression for $\omega_{1}$ is given in Equation (Equation 61) and $\kappa_{1},\;\kappa_{2}$ and $\omega_{2}$ are free parameters. In addition, $C_{2}$ and $C_{2}^{*}$ are constants of integration. The solution of Equation (Equation 1), rewritten in its primary form is: (67)$$\begin{array}{c} u\left(x,t\right)=3-[\frac{1}{2}+\frac{\sqrt{33}}{6}\tanh\left[\frac{\sqrt{33}}{6}\mu_{2}\left(\kappa_{1}x+\omega_{1}t\right)\right]-e^{\frac{\sqrt{33}}{6}\mu_{2}\left(\kappa_{1}x+\omega_{1}t\right)}/\\ 2\cosh\left[\frac{\sqrt{33}}{6}\mu_{2}\left(\kappa_{1}x+\omega_{1}t\right)\right]\left(\frac{\sqrt{33}}{11}+2C_{2}e^{\frac{\sqrt{33}}{6}\mu_{2}\left(\kappa_{1}x+\omega_{1}t\right)}\cosh\left[\frac{\sqrt{33}}{6}\mu_{2}\left(\kappa_{1}x+\omega_{1}t\right)\right]\right)]\\ -[\frac{1}{2}+\frac{\sqrt{33}}{6}\tanh\left[\frac{\sqrt{33}}{6}\mu_{2}\left(\kappa_{1}x+\omega_{1}t\right)\right]-e^{\frac{\sqrt{33}}{6}\mu_{2}\left(\kappa_{1}x+\omega_{1}t\right)}/\\ 2\cosh\left[\frac{\sqrt{33}}{6}\mu_{2}\left(\kappa_{1}x+\omega_{1}t\right)\right]\left(\frac{\sqrt{33}}{11}+2C_{2}e^{\frac{\sqrt{33}}{6}\mu_{2}\left(\kappa_{1}x+\omega_{1}t\right)}\cosh\left[\frac{\sqrt{33}}{6}\mu_{2}\left(\kappa_{1}x+\omega_{1}t\right)\right]\right){]}^{2}\\ -[\frac{1}{2}+\frac{\sqrt{33}}{6}\tanh\left[\frac{\sqrt{33}}{6}\nu_{2}\left(\kappa_{2}x+\omega_{2}t\right)\right]-e^{\frac{\sqrt{33}}{6}\nu_{2}\left(\kappa_{2}x+\omega_{2}t\right)}/\\ 2\cosh\left[\frac{\sqrt{33}}{6}\nu_{2}\left(\kappa_{2}x+\omega_{2}t\right)\right]\left(\frac{\sqrt{33}}{11}+2C_{2}^{*}e^{\frac{\sqrt{33}}{6}\nu_{2}\left(\kappa_{2}x+\omega_{2}t\right)}\cosh\left[\frac{\sqrt{33}}{6}\nu_{2}\left(\kappa_{2}x+\omega_{2}t\right)\right]\right)]\\ -[\frac{1}{2}+\frac{\sqrt{33}}{6}\tanh\left[\frac{\sqrt{33}}{6}\nu_{2}\left(\kappa_{2}x+\omega_{2}t\right)\right]-e^{\frac{\sqrt{33}}{6}\nu_{2}\left(\kappa_{2}x+\omega_{2}t\right)}/\\ 2\cosh\left[\frac{\sqrt{33}}{6}\nu_{2}\left(\kappa_{2}x+\omega_{2}t\right)\right]\left(\frac{\sqrt{33}}{11}+2C_{2}^{*}e^{\frac{\sqrt{33}}{6}\nu_{2}\left(\kappa_{2}x+\omega_{2}t\right)}\cosh\left[\frac{\sqrt{33}}{6}\nu_{2}\left(\kappa_{2}x+\omega_{2}t\right)\right]\right){]}^{2} \end{array}$$

For the particular case $\mu_{1}=0$ and $\nu_{0}=0$, the simple equations reduce to the following form:(68)$$\frac{df_{1}}{d\xi_{1}}=\mu_{0}+\mu_{2}f_{1}^{2},\;\;\frac{df_{2}}{d\xi_{2}}=\nu_{1}f_{2}+\nu_{2}f_{2}^{2},$$

In this case, one simple non-trivial solution of the system (Equation 27) is:(69)$$\begin{array}{c} \zeta_{0}=\eta_{0}=0,\;\zeta_{1}=-30\;\frac{\kappa_{1}+\kappa_{2}}{\alpha },\;\zeta_{2}=30\;\frac{\kappa_{1}+\kappa_{2}}{\alpha },\;\eta_{1}=\eta_{2}=30\;\frac{\kappa_{1}+\kappa_{2}}{\alpha },\\ \mu_{0}=1,\;\mu_{2}=-1,\nu_{1}=-1,\;\nu_{2}=1,\;\beta =-\alpha \left(\kappa_{1}+\kappa_{2}\right),\;\gamma =\frac{1}{5}\;\left({\kappa_{1}}^{2}+{\kappa_{2}}^{2}\right)\alpha^{2},\\ \delta =\frac{1}{2}\left(\kappa_{1}+\kappa_{2}\right)\left(2\alpha -10\left(\kappa_{1}+\kappa_{2}\right)\right),\;\epsilon =-\frac{1}{5}\left(\kappa_{1}^{2}+\kappa_{2}^{2}\right)\left(2\alpha -10\left(\kappa_{1}+\kappa_{2}\right)\right) \end{array}$$
where $\alpha ,\kappa_{1},\kappa_{2},\omega_{1},\omega_{2}$ are free parameters. The solutions of Equation (Equation 68), presented by the special functions *V* are:(70)$$f_{1}=V_{1,0,-1}\left(\xi_{1};1,1,2\right),\;\;f_{2}=V_{0,-1,1}\left(\xi_{2};1,1,2\right)$$

Then, the solution of Equation (Equation 59) transforms to
(71)$$\begin{array}{c} u\left(\xi_{1},\xi_{2}\right)=-30\;\frac{\kappa_{1}+\kappa_{2}}{\alpha }V_{1,0,-1}\left(\xi_{1};1,1,2\right)+30\;\frac{\kappa_{1}+\kappa_{2}}{\alpha }V_{1,0,-1}^{2}\left(\xi_{1};1,1,2\right)\\ +30\;\frac{\kappa_{1}+\kappa_{2}}{\alpha }V_{0,1,-1}\left(\xi_{2};1,1,2\right)+30\;\frac{\kappa_{1}+\kappa_{2}}{\alpha }V_{0,1,-1}^{2}\left(\xi_{2};1,1,2\right), \end{array}$$
where in the context of the solution of the extended tanh-function equation (see Equation (Equation 19)) and the solution of the Bernoulli equation (see Equation (Equation 12)), the solutions of Equation (Equation 68) obtain the following form:(72)$$V_{1,0,1}\left(\xi_{1};1,1,2\right)=\tanh\left(\xi_{1}\right),\;\;V_{0,-1,1}\left(\xi_{2};1,1,2\right)=\frac{1}{1+\exp(\xi_{2})},$$
where $\tanh(\xi_{1})<1$. In Equation (Equation 72), $\xi_{1}=\kappa_{1}x+\omega_{1}t$ and $\xi_{2}=\kappa_{2}x+\omega_{2}t$, as $\kappa_{1},\;\kappa_{2},\omega_{1}$ and $\omega_{2}$ are free parameters.

Then, for this simplest case, the final form of Equation (Equation 1) is:(73)$$\begin{array}{c} u\left(x,t\right)=-30\;\frac{\kappa_{1}+\kappa_{2}}{\alpha }\tanh\left(\kappa_{1}x+\omega_{1}t\right)+30\;\frac{\kappa_{1}+\kappa_{2}}{\alpha }\tanh^{2}\left(\kappa_{1}x+\omega_{1}t\right)\\ +30\;\frac{\kappa_{1}+\kappa_{2}}{\alpha }\frac{1}{1+\exp(\kappa_{2}x+\omega_{2}t)}+30\;\frac{\kappa_{1}+\kappa_{2}}{\alpha }{\left(\frac{1}{1+\exp(\kappa_{2}x+\omega_{2}t)}\right)}^{2} \end{array}$$

### 3.4. Case $m_{1}=1,\;m_{2}=1$

For this case, according to the balance Equation (Equation 23), $n_{1}=n_{2}=0$. However, for the simple equations
(74)$$\frac{df_{1}}{d\xi_{1}}=\mu_{1}f_{1},\;\;\;\;\frac{df_{2}}{d\xi_{2}}=\nu_{1}f_{2},$$
the general solution of Equation (Equation 1) can be presented in the following specific form:(75)$$u\left(\xi_{1},\xi_{2}\right)=1+f_{1}\left(\xi_{1}\right)f_{2}\left(\xi_{2}\right)$$The substitution of Equations (Equation 74) and (Equation 75) in Equation (Equation 22) leads to the following system of non-linear algebraic equations:(76)$$\begin{array}{c} \gamma \;\kappa_{2}\nu_{1}+\gamma \;\kappa_{1}\mu_{1}+\gamma \;\kappa_{1}\nu_{1}+\gamma \;\kappa_{2}\mu_{1}=0\\ \epsilon \;\kappa_{2}\nu_{1}+2\;\gamma \;\kappa_{2}\nu_{1}+\epsilon \;\kappa_{1}\mu_{1}+\beta \;{\kappa_{2}}^{3}{\mu_{1}}^{3}+\beta \;{\kappa_{1}}^{3}{\nu_{1}}^{3}-\alpha \;{\kappa_{2}}^{4}{\nu_{1}}^{3}+\epsilon \;\kappa_{2}\mu_{1}\\ +\beta \;{\kappa_{1}}^{3}{\mu_{1}}^{3}+2\;\gamma \;\kappa_{1}\mu_{1}+\beta \;{\kappa_{2}}^{3}{\nu_{1}}^{3}-\alpha \;{\kappa_{1}}^{4}{\mu_{1}}^{3}-\alpha \;{\kappa_{1}}^{4}{\nu_{1}}^{3}\\ +\epsilon \;\kappa_{1}\nu_{1}+2\;\gamma \;\kappa_{1}\nu_{1}+2\;\gamma \;\kappa_{2}\mu_{1}-3\;\alpha \;{\kappa_{1}}^{4}\mu_{1}{\nu_{1}}^{2}+3\;\beta \;{\kappa_{1}}^{3}{\mu_{1}}^{2}\nu_{1}\\ +3\;\beta \;{\kappa_{1}}^{3}\mu_{1}{\nu_{1}}^{2}-3\;\alpha \;{\kappa_{2}}^{4}{\mu_{1}}^{2}\nu_{1}-3\;\alpha \;{\kappa_{2}}^{4}\mu_{1}{\nu_{1}}^{2}+3\;\beta \;{\kappa_{2}}^{3}{\mu_{1}}^{2}\nu_{1}\\ +3\;\beta \;{\kappa_{2}}^{3}\mu_{1}{\nu_{1}}^{2}-\alpha \;{\kappa_{2}}^{4}{\mu_{1}}^{3}-3\;\alpha \;{\kappa_{1}}^{4}{\mu_{1}}^{2}\nu_{1}=0\\ \gamma \;\kappa_{2}\nu_{1}+\gamma \;\kappa_{1}\nu_{1}+\gamma \;\kappa_{1}\mu_{1}+\gamma \;\kappa_{2}\mu_{1}+5\;{\kappa_{2}}^{5}{\mu_{1}}^{4}\nu_{1}+{\kappa_{1}}^{5}{\nu_{1}}^{5}+\omega_{1}\nu_{1}+\omega_{2}\nu_{1}\\ +{\kappa_{1}}^{5}{\mu_{1}}^{5}+10\;{\kappa_{2}}^{5}{\mu_{1}}^{3}{\nu_{1}}^{2}+10\;{\kappa_{1}}^{5}{\mu_{1}}^{2}{\nu_{1}}^{3}+5\;{\kappa_{1}}^{5}\mu_{1}{\nu_{1}}^{4}+10\;{\kappa_{1}}^{5}{\mu_{1}}^{3}{\nu_{1}}^{2}\\ +5\;{\kappa_{2}}^{5}\mu_{1}{\nu_{1}}^{4}+\delta \;{\kappa_{1}}^{3}{\nu_{1}}^{3}+10\;{\kappa_{2}}^{5}{\mu_{1}}^{2}{\nu_{1}}^{3}+5\;{\kappa_{1}}^{5}{\mu_{1}}^{4}\nu_{1}+\delta \;{\kappa_{2}}^{3}{\mu_{1}}^{3}\\ +\delta \;{\kappa_{1}}^{3}{\mu_{1}}^{3}+\delta \;{\kappa_{2}}^{3}{\nu_{1}}^{3}+{\kappa_{2}}^{5}{\mu_{1}}^{5}+{\kappa_{2}}^{5}{\nu_{1}}^{5}+3\;\delta \;{\kappa_{1}}^{3}{\mu_{1}}^{2}\nu_{1}+3\;\delta \;{\kappa_{1}}^{3}\mu_{1}{\nu_{1}}^{2}\\ +3\;\delta \;{\kappa_{2}}^{3}{\mu_{1}}^{2}\nu_{1}+3\;\delta \;{\kappa_{2}}^{3}\mu_{1}{\nu_{1}}^{2}+\omega_{2}\mu_{1}+\omega_{1}\mu_{1}-3\;\alpha \;{\kappa_{1}}^{4}\mu_{1}{\nu_{1}}^{2}\\ -3\;\alpha \;{\kappa_{2}}^{4}{\mu_{1}}^{2}\nu_{1}-3\;\alpha \;{\kappa_{2}}^{4}\mu_{1}{\nu_{1}}^{2}+\epsilon \;\kappa_{2}\nu_{1}-\alpha \;{\kappa_{2}}^{4}{\nu_{1}}^{3}+\epsilon \;\kappa_{1}\mu_{1}\\ +\epsilon \;\kappa_{2}\mu_{1}-\alpha \;{\kappa_{1}}^{4}{\mu_{1}}^{3}+\epsilon \;\kappa_{1}\nu_{1}-\alpha \;{\kappa_{1}}^{4}{\nu_{1}}^{3}-\alpha \;{\kappa_{2}}^{4}{\mu_{1}}^{3}-3\;\alpha \;{\kappa_{1}}^{4}{\mu_{1}}^{2}\nu_{1}=0 \end{array}$$

One non-trivial solution of the system (Equation 76) is:(77)$$\mu_{1}=-\nu_{1},\;\gamma =-\frac{1}{2}\epsilon ,$$
where $\alpha ,\beta ,\delta ,\epsilon ,\nu_{1},\kappa_{1},\kappa_{2},\omega_{1},\omega_{2}$ are free parameters. For this case, the solutions of Equation (Equation 74) can be presented as
(78)$$f_{1}=V_{0,-\nu_{1}}\left(\xi_{1};1,1,1\right)=\exp\left(-\nu_{1}\xi_{1}\right),\;\;f_{2}=V_{0,\nu_{1}}\left(\xi_{2};1,1,1\right)=\exp\left(\nu_{1}\xi_{2}\right)$$Then, the solution of Equation (Equation 1) becomes:(79)$$u\left(\xi_{1},\xi_{2}\right)=1+V_{0,-\nu_{1}}\left(\xi_{1};,1,1,1\right)V_{0,\nu_{1}}\left(\xi_{2};,1,1,1\right),$$
or
(80)$$u\left(\xi_{1},\xi_{2}\right)=1+\exp\left[-\nu_{1}\left(\kappa_{1}x+\omega_{1}t\right)\right]\exp\left[\nu_{1}\left(\kappa_{2}x+\omega_{2}t\right)\right]$$

Several illustrative numerical examples of one analytical solution of Equation (Equation 1) obtained in this study are presented in Figure 1, Figure 2 and Figure 3. As can be seen, different complex multi-soliton structures can be observed depending on the numerical values the free parameters in Equation (Equation 36) as well depending on the numerical space and time intervals chosen for the simulations. In this specific case, we vary only the numerical value of the travelling-wave velocity $\omega_{2}$, as the numerical value of the travelling-wave velocity $\omega_{1}$ varies indirectly, too (see Equation (Equation 28)). We also vary the space coordinate in the numerical intervals from [−10,10] to [−100,100], while the time intervals vary from [0,0.03] to [0,1.5].

## 4. Conclusions

In this paper, we have shown the effectiveness of the SEsM for obtaining exact solutions of a famous evolution equation in mathematical physics. We have presented various types of the travelling-wave solution of the fifth-order KdV equation, using the special functions *V*, which are solutions of so-called simple equations in SEsM. The obtained results are only a part of the possible variety of exact solutions of the studied equation that can be derived using the special functions *V*. We believe that the presented results are new. Moreover, the use of composite functions in the methodology of the SEsM gives possibilities for obtaining other specific solutions of the physical phenomena, discussed in the paper. However, this will be the goal of further investigations.

## Figures and Tables

**Figure 1 entropy-24-01288-f001:**
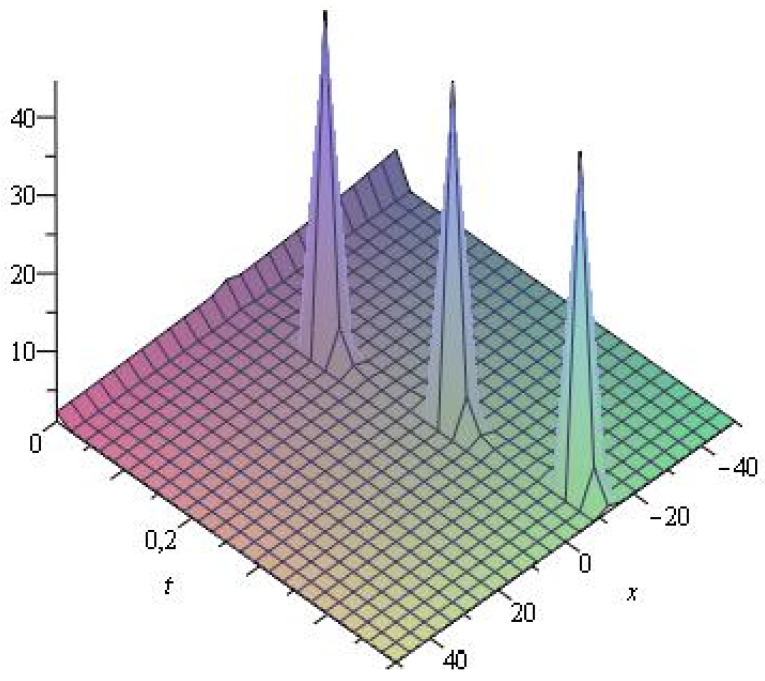
Numerical simulation of Equation (Equation 36) for $\mu_{1}=2,\nu_{1}=3,\kappa_{1}=0.001,\kappa_{2}=0.02,\omega_{2}=0.6$.

**Figure 2 entropy-24-01288-f002:**
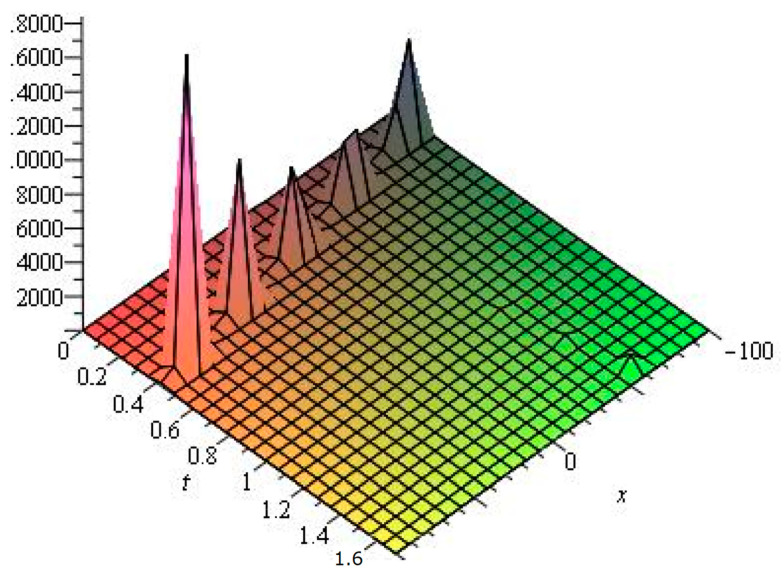
Numerical simulation of Equation (Equation 36) for $\mu_{1}=2,\nu_{1}=3,\kappa_{1}=0.001,\kappa_{2}=0.02,\omega_{2}=1$.

**Figure 3 entropy-24-01288-f003:**
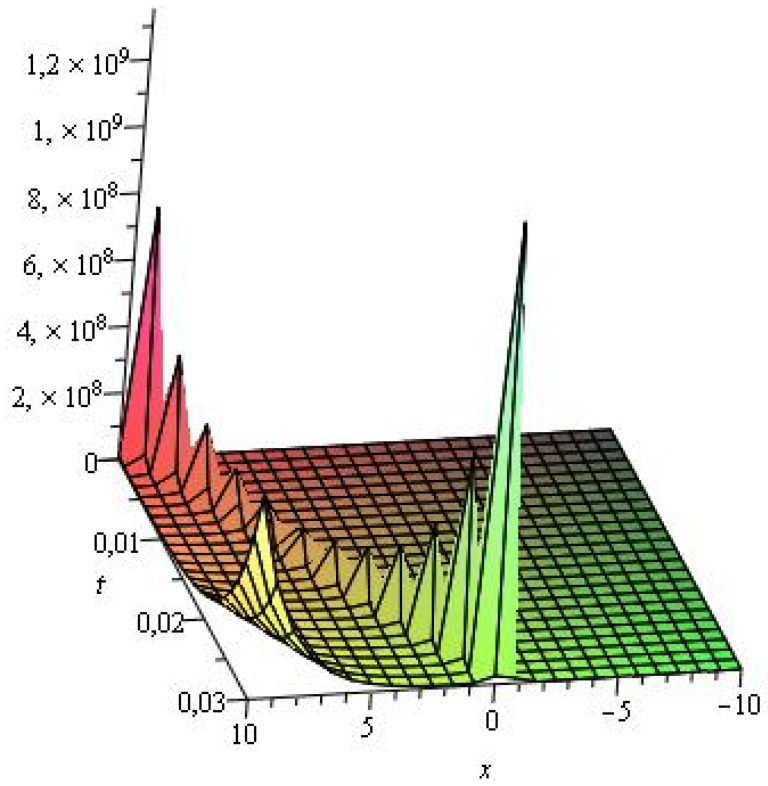
Numerical simulation of Equation (Equation 36) for $\mu_{1}=2,\nu_{1}=3,\kappa_{1}=0.001,\kappa_{2}=0.02,\omega_{2}=5$.

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
