# Peer review of "Exact Travelling-Wave Solutions of the Extended Fifth-Order Korteweg-de Vries Equation via Simple Equations Method (SEsM): The Case of Two Simple Equations"

_entropy, 2022, doi:10.3390/e24091288_

Round 1

Reviewer 1 Report

Report on the paper

Exact Travelling Wave Solutions of the Extended Fifth-order Korteweg - deVries Equation via Simple Equations Method (SEsM): The Case of Two Simple Equations

by Elena V. Nikolova

The authors' aim is to 'apply the Simple Equations Method (SEsM) for obtaining exact travelling wave solutions of the extended fifth–order Korteweg - deVries (KdV) equation'. 

The basic idea is to consider the solution of this equation as a composite function of two functions of two independent variables.

The calculations seem correct to me and the results sound well. 

The content is valuable and I recommend the publication of the work.

Author Response

Thank you for your review.

With best wishes,

Elena Nikolova

Reviewer 2 Report

The paper is devoted to the search of exact solutions of the extended Korteweg – deVries equation. Author explains the essence of the Simple Equations Methods, gives the background of this method and demonstrates it for the extended KdV equation in the case of two simple equations with various powers (1,2 or 3) of the unknown function in them. I have amendments for the text.

1. Bringing such a huge number of references (the first 35 references) to confirm the obvious statements about the importance of differential equations for modeling the processes of the surrounding world looks very strange. Until you see that 17 of these 35 links are to the author's own works. I think it's enough to leave 5-6 links. It would be better if, in this context, these would be references not to their own articles, but to significant, well-known monographs by various authors.

2.       In general, 145 references to 30 pages of text looks like a bust.

3.       After unprecedented thoroughness in writing out algebraic equations that completely cover pages 8-11, I would like to see more detailed reasoning when choosing a non-trivial solution (28). Why this particular decision? For what reasons is the number of free parameters chosen? And what will other possible solutions give? 

Author Response

Dear reviewer,

Thank you for your pertinent recommendations. Based on them, the following corrections of the manuscript have been made:

1.-2. The biblography of the manuscript has been reduced from 145 references to 94 references. References connected to the importance of use of differential equations is reduced to 4 monographs. All auto-citations are removed.

3. As you know, obtaing solutions of a NPDE with hight order is quite a laborious process even though computational software. This is done step by step starting from the simplest equation of the system of algebraic equations and then moving in direction of solving more coplicated equations. There are many solutions, which can be obtained by the computational software, but our goal is ro express the coefficients of the solved equation and the coefficients of its solution by the coefficents of the simple equations and the coefficients of the solutions of the simple equations so far as it is possible. This is the main reason for our choice of free parameters in the calculation process and in the obtaned solutions of the solved equation, respectively, if it is necessary. In some of the examples, considered in this study, however, dsspite the large number algebraic equations of the system (27), the calculation process is very difficult becuse of the complicated and the long expessions resulting from the hight degree of Eq. (1). Therefore, when calculating the solutions of Eqs (27), we take the simplest roots of some of its variables, such as 0 or 1. For some cases considered in the study, we choose these variables to be the coefficients of the solved equation, keeping the coefficents in front of the highest degree of the corresponding solution. In other cases, resetting of some coefficients of the simple equations are necessary to reach the form of the corresponding simle equations with known analytical solutions. (The simple ODEs with known analytical solutions are presented in the end of Sec. 2 of the manuscipt.).

All corrections are marked in red in the manuscript.

With best wishes,

Elena Nikolova
